# Managing Irinotecan-Induced Diarrhea: A Comprehensive Review of Therapeutic Interventions in Cancer Treatment

**DOI:** 10.3390/ph18030359

**Published:** 2025-03-02

**Authors:** Xiaoqin Yang, Jiamei Chen, Yitao Wang, Yihan Wu, Jinming Zhang

**Affiliations:** 1State Key Laboratory of Southwestern Chinese Medicine Resources, School of Pharmacy, Chengdu University of Traditional Chinese Medicine, Chengdu 611137, China; yangxiaoqin202033@126.com (X.Y.); cobby0945@163.com (J.C.); 2State Key Laboratory of Quality Research in Traditional Chinese Medicine, Institute of Chinese Medical Sciences, University of Macau, Macau SAR 999078, China; ytwang@umac.mo

**Keywords:** CPT-11(irinotecan), SN-38, CPT-11-induced diarrhea (CID), diarrhea prevention, clinical treatment

## Abstract

Irinotecan (CPT-11), an inhibitor of DNA topoisomerase I, stands as a pivotal therapeutic agent in oncology. However, its use is primarily constrained by side effects such as neutropenia and the onset of delayed diarrhea. Despite the effective management of neutropenia, CPT-11-induced diarrhea (CID) is often severe, leading to hospitalization, dosage adjustments, and in some cases, treatment discontinuation, which can significantly impact therapeutic outcomes. A multitude of pharmacological agents have been investigated in preclinical and clinical studies with the aim of reducing or preventing the onset of delayed diarrhea associated with CPT-11. This comprehensive review examines the underlying mechanisms of CPT-11-triggered delayed diarrhea and discusses the experimental medications and strategies that have been utilized to combat this adverse effect. This review encompasses an exploration of chemical formulations, the application of traditional Chinese medicine, and the advent of innovative drug delivery systems. It is anticipated that this article will serve as a valuable resource for both novice researchers in the realm of irinotecan chemotherapy and for those who are well-versed in the field, including experts and practicing clinicians.

## 1. Introduction

Irinotecan (CPT-11), the inaugural representative of topoisomerase I inhibitors extracted from the camptotheca acuminata plant, has garnered acclaim for its broad-spectrum efficacy against various cancers, including colorectal, lung, breast, and malignant lymphoma. Its clinical prominence as an antineoplastic agent, however, is tempered by the significant adverse effects, most notably diarrhea, which have impeded its widespread adoption in oncology [1,2]. Diarrhea, a prominent and dose-limiting toxicity associated with CPT-11, manifests in two forms: acute and delayed. Acute diarrhea, often observed early in the course of CPT-11 administration, is linked to the inhibitory effect of drugs on cholinergic activity. This inhibition leads to an accumulation of acetylcholine, which in turn increases intestinal motility, disrupts absorption, and provokes contractility—effects that can be mitigated with anticholinergic agents such as atropine [3]. In contrast, delayed diarrhea, which presents over 24 h post-administration, is more severe and persistent, and is often challenging to manage. Characterized by its non-cumulative nature, it affects up to 87% of patients, with 30–40% experiencing severe grade 3 or 4 diarrhea [4,5]. Prolonged diarrhea can lead to renal insufficiency and electrolyte disturbances due to fluid and electrolyte losses and may provoke cardiovascular complications as a result of intravascular volume fluctuations. These complications can necessitate the cessation of CPT-11 treatment and pose life-threatening risks. Furthermore, severe diarrhea significantly diminishes patients’ quality of life, disrupting their social roles, interpersonal relationships, and leading to a sense of social isolation.

Indeed, the pathophysiological mechanisms underlying delayed diarrhea induced by CPT-11 have garnered significant interest and have been the focus of extensive experimental research. The severity of this iatrogenic condition is predominantly attributed to the accumulation of the active metabolite, SN-38, derived from CPT-11. The metabolic conversion of CPT-11 to SN-38 is facilitated by carboxylesterases, specifically CES1 and CES2 [6,7]. Thereafter, SN-38 undergoes further metabolism to SN-38G by the enzyme uridine diphosphate glucuronosyltransferase 1A1 (UGT1A1) [8,9]. SN-38G is excreted into the bile, where it can be acted upon by β-glucuronidase (β-GD) present in the intestinal lumen, leading to the release and accumulation of SN-38, which is a primary culprit in the etiology of CID [10]. Despite this, the precise mechanisms of SN-38-induced diarrhea remain a subject of ongoing debate. Additional research has highlighted that CPT-11 can alter the intestinal luminal environment, potentially promoting the proliferation of diverse bacterial species. The escalating activity of bacterial β-glucuronidase can deconjugate SN-38G, reinstating the active SN-38 form, which can precipitate severe intestinal damage and consequent diarrhea. Furthermore, CPT-11 has been shown to cause significant colonic damage, characterized by increased apoptosis, crypt hypoplasia, and dilation, as well as excessive mucus secretion and damage to the small intestine. These histopathological changes, coupled with alterations in the number of goblet cells, contribute to the development of diarrhea.

To date, a myriad of therapeutic strategies and pharmacological agents have been deployed to address the challenge of delayed diarrhea induced by CPT-11. These encompass the inhibition of SN-38 production, enhancement of SN-38 adsorption, and suppression of β-glucuronidase (β-GD) activity. Modern medicinal approaches include the use of loperamide, a synthetic opioid that moderates intestinal motility, extends transit time, and promotes fluid reabsorption, as well as its role in inhibiting the expression of TX-A2, thereby exerting an antisecretory effect [11]. Atropine, a competitive antagonist of muscarinic acetylcholine receptors, and targeted antidiarrheal therapies such as the somatostatin analog octreotide and nonsteroidal anti-inflammatory drugs (NSAIDs) are also utilized clinically to mitigate CID [12,13]. Furthermore, the rich tapestry of traditional Chinese medicine has been harnessed, with herbal formulas, extracts, and phytochemical substances being applied to alleviate diarrheal symptoms. Notable among these are Huangqin Decoction, Hange-Shashin-To, Sairei-To, Shengjiang Xiexin Decoction, and Gegen Qinlian Decoction, which have demonstrated efficacy in reducing diarrhea associated with CPT-11. The quest to enhance the management of CID continues to grow, with a keen interest in developing novel and effective treatment modalities to bolster therapeutic success.

This comprehensive review examines the underlying mechanisms of CPT-11-triggered delayed diarrhea and discusses the experimental medications and strategies that have been utilized to combat this adverse effect. The review encompasses an exploration of chemical formulations, the application of traditional Chinese medicine, and the advent of innovative drug delivery systems. We aim to provide a balanced overview of all major therapeutic interventions, highlighting their strengths and limitations. We acknowledge that the mechanisms and treatments for diarrhea discussed in this review represent just the beginning of a much larger body of knowledge. As biotechnology and genetic research continue to make groundbreaking strides, it is plausible that future developments will yield even more potent methods for managing recalcitrant chemotherapy-induced diarrhea. These advancements will not only enhance our understanding of the condition but also contribute significantly to the improvement of cancer treatment with chemotherapeutic agents.

## 2. Mechanisms of CPT-11-Induced Diarrhea

### 2.1. Mechanism of Pharmacokinetic

The metabolic pathway of CPT-11 is intricate, involving a multitude of enzymes responsible for drug metabolism. CPT-11, functioning as a prodrug, undergoes biotransformation into its active cellular metabolite, SN-38 (7-ethyl-10-hydroxycamptothecin), via the ester bond at the C-10 position. SN-38 demonstrates cytotoxic activity that is 100 to 1000 times more potent than that of CPT-11 itself [6,7]. Over recent years, the etiology of intestinal toxicity has been closely linked to the metabolic processes of CPT-11 and its active metabolite, SN-38, within the body. Figure 1 provides an overview of the metabolism of irinotecan, highlighting the key metabolites and enzymes involved in the onset of delayed diarrhea.

Upon intravenous administration, CPT-11 is rapidly bioactivated to SN-38 in the plasma and liver. This initial conversion is predominantly facilitated by carboxylesterases (CEs), which are also found in the small intestine, liver, and colon. The expression levels and activity of these CEs can significantly influence the balance between CPT-11 and SN-38, subsequently impacting the circulating concentrations and antineoplastic efficacy of SN-38 [14,15]. Additionally, CPT-11 can be metabolized into 7-ethyl-10-[4-N-(5-aminopentanoic acid)-1-piperidino]-carbonyloxy-camptothecine (APC) and 7-ethyl-10-(4-amino-1-piperidino)-carbonyloxy-camptothecine (NPC) by the cytochrome P450 enzyme system. Emerging research indicates that NPC serves as a substrate for carboxylesterase, being nearly entirely converted to SN-38, albeit with a minimal impact on the overall antitumor efficacy due to the scarce quantity of NPC produced [16,17].

Prior to its excretion into the bile, SN-38 is inactivated by conversion to SN-38-glucuronide (SN-38G) through the action of glucuronyl transferases (UGTs). Irinotecan and its metabolites, including SN-38, SN-38G, and APC, are then secreted via the bile into the small intestine, where they may undergo enterohepatic recirculation back to the liver. This process is mediated by transport proteins such as multidrug resistance-associated protein 1 (MDR1/ABCB1), multidrug resistance-associated protein 2 (MRP2/ABCC2), and breast cancer resistance protein (BCRP/ABCG2). Moreover, SN-38G can be reconverted to active SN-38 by β-glucuronidase, an enzyme produced by the intestinal bacterial flora. Concurrently, CPT-11 is re-hydrolyzed to SN-38 by CEs, allowing for the reabsorption of both CPT-11 and SN-38 through the intestinal lining. These compounds subsequently enter the bloodstream and are transported to the liver under the auspices of organic anion transporter polypeptides (OATPs), thereby completing the hepatic–intestinal cycle [18]. Figure 2 provides a detailed overview of the metabolic pathway of irinotecan, highlighting the key compounds and enzymes involved in its metabolism and the formation of active metabolites such as SN-38.

### 2.2. CPT-11 Factors Leading to Diarrhea

#### 2.2.1. The Effect of UGT1A1 Polymorphisms

Under the catalytic influence of carboxylesterase, CPT-11 is converted into its active metabolite, SN-38 [6]. Subsequently, SN-38 is metabolized into an inactive form, glucuronidated SN-38 (SN-38G), by the liver enzyme UGT1A1 [19]. These metabolites are ultimately excreted via the bile and eliminated from the body through feces. Currently, UGT1A1 is recognized as a critical determinant of SN-38 concentrations, which in turn influence the severity of intestinal toxicity [20]. There is a well-documented correlation between UGT1A1 gene polymorphisms and the toxicity associated with CPT-11 use both domestically and internationally.

UGT1A1 exhibits a high degree of polymorphism, with over 100 genetic variants identified to date [21]. The most pertinent UGT1A1 polymorphisms in relation to the pharmacokinetics and pharmacodynamics of CPT-11 are UGT1A1*6 and UGT1A1*28 [22]. The promoter region of the UGT1A1 gene features an atypical TATA domain comprising five to eight thymine–adenine (TA) repeats, with the six-repeat genotype being the most prevalent. An increase in the number of TA repeats is associated with reduced UGT1A1 expression. Notably, the UGT1A1*28 variant incorporates an additional TA repeat, which significantly diminishes UGT1A1 transcription and expression by approximately 70%, leading to decreased levels of SN-38G [19]. This genetic variant is more prevalent among Caucasians and Africans/African Americans, yet it is less common in the Asian population [23]. Individuals with the UGT1A1*28 non-wild type genotype exhibit a significantly higher incidence of diarrhea compared to those with the wild type, emphasizing the utility of gene polymorphism screening prior to CPT-11 chemotherapy to identify high-risk groups and anticipate potential adverse effects, thereby informing clinical decision-making [24].

Given the lower prevalence of UGT1A1*28 in Asians, its impact on toxicity outcomes is less pronounced in this demographic. Several meta-analyses have indicated that the UGT1A16 polymorphism may serve as a potential biomarker for predicting CPT-11-related toxicity in the Asian population [20,25]. Both Caucasian and Asian patients who are homozygous or heterozygous for the UGT1A1*28 variant are at an elevated risk for severe diarrhea following CPT-11 administration compared to wild-type patients, with a dose-dependent effect observed in a meta-analysis of Caucasian carriers [26]. In Asian patients, the UGT1A1*6 polymorphism is closely linked to the risk of CPT-11-induced neutropenia and is also significantly associated with severe diarrhea [20,27]. However, the dose dependency of this association remains unclear, as dose-subgroup analyses have not been conducted [27].

Beyond UGT1A1*6 and UGT1A1*28, other UGT1A1 polymorphisms may theoretically impact CPT-11-related toxicity [28]. For instance, UGT1A1*60, which is in linkage with UGT1A1*28, is associated with reduced transcriptional activity [29]. Nevertheless, UGT1A1*60 status has not been significantly correlated with CPT-11-related toxicities or pharmacokinetics in clinical studies [30]. Similarly, UGT1A1*93, also in linkage disequilibrium with UGT1A1*28, has been associated with increased SN-38 area under the curve (AUC), reduced neutrophil counts, hematological toxicity, diarrhea, and grade 3 vomiting [31].

Genetic polymorphisms in UGT1A9 and UGT1A7 are likewise intimately connected with the risk of diarrhea. Individuals with the UGT1A9*22 genotype demonstrate higher enzyme expression and increased SN-38 glucuronidation compared to those with UGT1A9*1/*1, thus facing a higher risk of diarrhea [32,33]. Conversely, the UGT1A7*3 and UGT1A7*4 polymorphisms are characterized by diminished enzyme activity and SN-38 binding, with UGT1A7*3/*3 carriers being at a greater risk for adverse events during CPT-11 chemotherapy [34].

#### 2.2.2. The Effect of Drug Transporter Polymorphisms

The ATP-binding cassette (ABC) membrane transporters play a pivotal role in the multidrug resistance observed in tumor cells. Members of the ABC family, including ABCB1, ABCC1, ABCC2, and ABCG2, are integral to the transport pathway of drugs within the body [35]. Given that both CPT-11 and its active metabolite, SN-38, are substrates for ABC transporters, polymorphisms in these transporter genes may significantly influence the pharmacokinetics and toxicity profiles of CPT-11. Clinical analyses have established a clear link between ABCB1 polymorphisms (specifically SNPs 1128503, rs2032582, and rs1045642) and the toxicities associated with CPT-11 treatment [36]. Individuals harboring these SNPs have been shown to exhibit a poorer response to CPT-11-based therapies and a shorter survival rate in advanced colorectal cancer [37].

Multivariate analyses have revealed associations between ABCC1 single nucleotide polymorphisms (SNPs rs6498588 and rs1750133) and increased plasma concentrations of SN-38, as well as decreased absolute neutrophil counts [38]. Conversely, the ABCB1 variant rs12720066 has been associated with reduced SN-38 exposure and elevated neutrophil levels. Beyond ABCB1 and ABCC1, polymorphisms in ABCC2 (rs3740066) and ABCG2 (rs2231137) have been identified as independent predictors of toxicity [34]. However, the impact of the ABCG2 (421C > A) polymorphism on CPT-11 exposure appears to be more limited [39]. The ABCC2 gene, which encodes an extrahepatic transporter, may confer a protective effect against diarrhea, potentially through reduced hepatobiliary transport of CPT-11, thereby lessening its intestinal exposure [40]. Although their specific role in CPT-11 efflux remains to be determined, ABCC5 and ABCG1 may also participate in this process, as several SNPs associated with these transporters have been linked to severe diarrhea [41].

Within the OATP family of genes, OATP1B1, encoded by the SLCO1B1 gene, is noted for its high uptake of SN-38 in the liver. Mutations in the SLCO1B1 gene are hypothesized to impact the transport activity of OATP1B1, consequently affecting the hepatic clearance of SN-38. Studies have highlighted that two prevalent SLCO1B1 gene mutations can influence the transport and metabolism of CPT-11, albeit through distinct mechanisms [42]. The SLCO1B1 (T521C) mutation predominantly reduces the affinity of OATP1B1 for its substrate, thereby impairing its transport efficiency. In contrast, the SLCO1B1 (A388G) mutation predominantly leads to diminished expression levels of OATP1B1, compromising its transport capacity and impacting the hepatic metabolism of SN-38, which can result in more severe adverse reactions [43]. Clinical observations have indicated that patients with the SLCO1B1**15 allele are at a significantly higher risk of experiencing diarrhea and neutropenia in the week following chemotherapy. The increased toxicity of CPT-11 in these patients may be attributable to enhanced distribution and bioavailability conferred by the SLCO1B1**15 variant [44,45].

#### 2.2.3. The Effect of CYP3A Polymorphisms

The CYP3A enzyme, a member of the cytochrome P450 superfamily, is encoded by the CYP3A gene located on human chromosome 7. It plays a crucial role in the metabolism of numerous drugs, including CPT-11. Theoretical considerations suggest that diminished CYP3A activity or expression could reduce the synthesis of APC and NPC, shunting the metabolic pathway towards increased conversion of CPT-11 to SN-38 by carboxylesterase (CES). This metabolic shift could potentially elevate the risk of adverse reactions due to heightened SN-38 production. Consequently, polymorphisms in the CYP3A gene are hypothesized to be associated with the variability in CPT-11’s adverse reaction profiles [46].

A number of single nucleotide polymorphisms (SNPs) in the CYP3A4 gene have been documented, with their prevalence varying significantly across different ethnic groups. Notably, the CYP3A15 variant (485G > A [Arg162Gln]) is observed in 2–4% of African Americans, while other variants such as CYP3A2 (664T > C [Ser222Pro]), CYP3A10 (520G > C [Asp174His]), and CYP3A17 (566T > C [Phe189Ser]) are more prevalent among Caucasians and Mexicans, affecting 2–5% of individuals. In East Asian populations, the CYP3A16 (554C > G [Thr185Ser]) and CYP3A18 (878T > C [Leu293Pro]) variants are more commonly encountered, affecting 1–10% of individuals [47,48].

Research has indicated that variations in the CYP3A4 genotype can influence the clearance rate of CPT-11. However, the clinical relevance of CYP3A4 SNPs is often overshadowed by the low frequency of these variants and the substantial influence of exogenous and endogenous factors on enzyme activity. Inter-individual differences in enzyme activity are more likely to be attributable to environmental and physiological factors, including drug interactions, nutritional status, alterations in liver function, and the overall health condition of the patient, rather than genetic polymorphisms alone. As a result, the inclusion of CYP3A4 gene detection in routine clinical practice remains limited [49].

#### 2.2.4. Pathophysiology of CPT-11-Induced Diarrhea

The alteration of the intestinal milieu post-CPT-11 administration is a recognized mechanism that precipitates diarrhea [50]. Treatment with CPT-11 can lead to significant damage to both the colonic and small intestinal tissues. This damage is characterized by heightened apoptosis, a diminished villi-to-crypt ratio, dilated crypts, increased lymphatic infiltration within the mucosa, and excessive mucus secretion accompanied by villous atrophy [51]. Moreover, the active metabolite SN-38 induces direct mucosal injury, which manifests as malabsorption of water and electrolytes and heightened mucosal secretion [52].

Following CPT-11 administration, there are notable shifts in fecal sodium and potassium levels, prompting an osmotic flow of water into the intestinal lumen to restore electrolyte balance, a process that culminates in diarrhea. Concurrently, there are significant alterations in serum sodium, chloride, and osmolality levels. In vivo studies have shown that CPT-11 can result in a thinner intestinal wall and ileal epithelial vacuolization associated with apoptosis, further exacerbating malabsorption. Additionally, CPT-11 induces goblet cell hyperplasia and an excess of sulfomucin in the cecum, pointing to increased mucin secretion [53].

The epithelial barrier is crucial for intestinal function, and CPT-11 has been shown to compromise the integrity of tight junctions, including the vital protein components claudin-1 and occludin [50]. This disruption can lead to bacterial translocation and exacerbate diarrhea. Furthermore, CPT-11 induces alterations in the gut microbiota, which can contribute to chemotherapy-induced mucositis. The changed luminal environment can also promote the proliferation of bacteria capable of producing β-glucuronidase, an enzyme that can convert the inactive SN-38 glucuronide (SN-38G) back to the active SN-38, thereby augmenting intestinal damage and diarrhea [51].

The direct role of the intestinal microbiota in CID has been highlighted by studies such as those by Giovanni Brandi and colleagues [54]. Using germ-free and holoxenic mice, they demonstrated that the presence of a normal gut microbiota can significantly influence the severity of CID and intestinal epithelial damage. Notably, holoxenic mice treated with a lethal dose of CPT-11 exhibited severe intestinal mucosal damage, whereas germ-free mice showed no such damage, underscoring the role of bacterial factors in the pathogenesis of CID.

#### 2.2.5. The Dose-Dependent Relationship Between CPT-11 Dosage and Diarrhea Incidence

The incidence and severity of diarrhea associated with irinotecan (CPT-11) treatment have been well-documented in numerous clinical studies, highlighting a clear dose-dependent relationship. Higher doses of CPT-11 are consistently linked to increased frequency and severity of both acute and delayed diarrhea, posing significant challenges in clinical management.

In a phase II study by Shimada et al. [4], the administration of escalating doses of CPT-11 in patients with metastatic colorectal cancer revealed a direct correlation between dose intensity and the incidence of diarrhea. Specifically, patients receiving higher doses exhibited a significantly higher rate of grade 3 and 4 diarrheas compared to those on lower dosing regimens. This observation underscores the need for meticulous dose titration to balance therapeutic efficacy and adverse effects.

Further supporting this dose-dependent relationship, Abigerges et al. [11] demonstrated that high-dose CPT-11 regimens necessitated intensive management with high-dose loperamide to control severe diarrhea. Their findings indicated that the severity of diarrhea increased proportionally with higher doses of CPT-11, necessitating careful monitoring and intervention to prevent life-threatening complications.

Moreover, the pharmacokinetic and pharmacodynamic properties of CPT-11 have been extensively studied, revealing that its active metabolite, SN-38, is primarily responsible for the observed gastrointestinal toxicity [47]. The conversion of CPT-11 to SN-38 is facilitated by carboxylesterases, and the subsequent metabolism of SN-38 to its inactive form, SN-38G, is mediated by UGT1A1 [31]. Genetic polymorphisms in UGT1A1, such as UGT1A1*28, have been identified as significant predictors of increased SN-38 levels and, consequently, higher incidence of severe diarrhea [55]. This genetic variability further complicates the dose-dependent relationship, as individuals with certain UGT1A1 genotypes may be at higher risk for adverse events at standard doses.

In addition to genetic factors, clinical studies have shown that the timing and schedule of CPT-11 administration also influence the incidence of diarrhea. For instance, a study by JP Armand [5] demonstrated that the frequency of severe diarrhea was significantly higher in patients receiving CPT-11 every 3 weeks compared to those on a more frequent dosing schedule. This suggests that the dosing interval may modulate the accumulation and systemic exposure of SN-38, thereby affecting the severity of gastrointestinal toxicity.

In summary, the dose-dependent relationship between CPT-11 and diarrhea incidence is well-established through clinical and pharmacogenetic studies. Higher doses of CPT-11 are associated with increased frequency and severity of diarrhea, necessitating careful dose adjustment and patient monitoring. Future research should focus on personalized dosing strategies that account for genetic variability and pharmacokinetic profiles to optimize therapeutic outcomes while minimizing adverse effects.

**Figure 2 pharmaceuticals-18-00359-f002:**
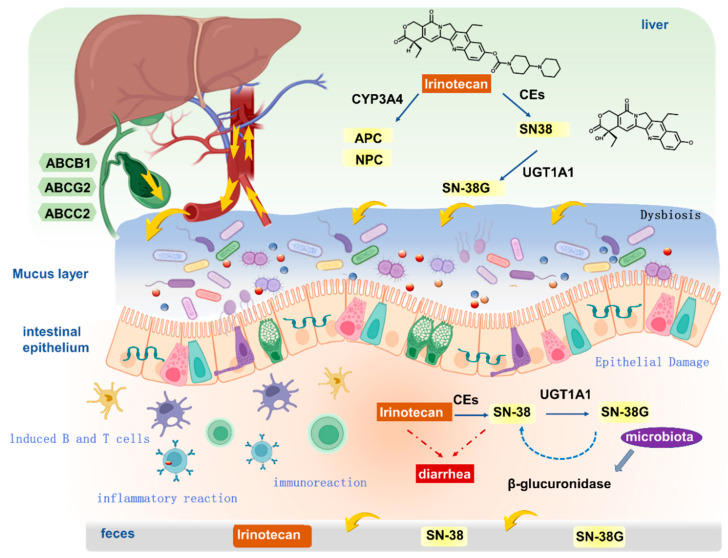
Overview of irinotecan metabolism. CPT-11 is a prodrug that is converted to active metabolite ethyl-10-hydroxy-camptothecin (SN-38) by liver carboxylesterase-converting enzymes (CES1/2) and is then transported to the liver by 1B1 polypeptide (OATP1B1) and inactivated by microsomal uridine 5′-diphospho-glucuronosyltransferase enzymes (UGTs): UGT1A1. Irinotecan is transported to the bile by a group of ATP-binding cassette transporters (ABC transporters): ABCB1, ABCC2, and ABCG2. Irinotecan is efficiently metabolized by cytochrome P450 enzymes: CYP3A4 and CYP3A5. This results in the generation of less active metabolites APC (7-ethyl-10-[4-N-(5-aminopentanoic acid)-1-piperidino] carbonyloxycamptothecin) and NPC (7-ethyl-10-[4-amino-1-piperidino] carbonyloxycamptothecin). NPC (but not APC) can be further converted to SN-38 by CES1, and CES2 gut microbiota may also participate in irinotecan metabolism by the production of β-glucuronidase, which catalyzes the breakdown of SN-38G into SN-38.

## 3. Strategies to Block or Treat Delayed Diarrhea

A spectrum of preventive and therapeutic strategies has been theorized and evaluated in both animal models and clinical studies, aimed at mitigating or counteracting CID [56]. The proactive approach of diarrhea prevention is widely regarded as the optimal strategy to avert this grave drug-related complication. Implementing such a strategy not only enhances the safety profile of the drug but also has the potential to decrease healthcare costs associated with hospitalization, elevate the quality of life for patients, and possibly facilitate escalated dosing regimens. This could, in turn, augment the therapeutic efficacy of the treatment through an improved tumor response [57]. Beyond the initial preventive steps endorsed by clinical guidelines, which encompass adjustments to the treatment schedule and dosage, as well as genetic screening, this review delves into the utilization of traditional Chinese medicine and modern pharmaceuticals. It specifically highlights their application in either forestalling or treating diarrhea that arises as a consequence of CPT-11 administration.

### 3.1. Chemical Drug Treatment of Delayed Diarrhea

#### 3.1.1. Antidiarrheal Therapy

In accordance with the current clinical guidelines, loperamide is widely acknowledged as the first-line therapeutic agent for the management of CID. Loperamide functions as an inhibitor of intestinal peristalsis by activating opioid receptors within the intestinal plexus and blocking the release of acetylcholine, thereby reducing intestinal motility and secretion [58]. Research has demonstrated that while diarrhea poses a dose-limiting toxicity at a dosage of 350 mg/m^2^ for CPT-11 administered every three weeks, the concurrent use of high-dose loperamide can potentially escalate the CPT-11 dosage to 750 mg/m^2^. Nonetheless, loperamide is not without side effects and has been noted to have a relatively high failure rate in treating CID [59].

Octreotide, a somatostatin analog and an inhibitor of intestinal secretion, is proposed as a second-line treatment for CID following the inadequacy of loperamide. It mitigates diarrhea by curbing gastrointestinal hormone secretion, diminishing intestinal peristalsis, extending gastrointestinal transit time, and enhancing the reabsorption of water and electrolytes while reducing secretion [60]. Although octreotide has shown promise in the secondary prevention of refractory CID in a limited case series, there is an acknowledged scarcity of robust research on its efficacy in the context of CID [61,62].

Acetorphan, an oral active enkephalinase inhibitor marketed as Racecadotril or Tiorfan^®^, has been investigated as an alternative to loperamide due to its antidiarrheal and antisecretory properties [63]. A low-dose escalation study indicated that prophylactic acetylmorphine could significantly reduce the incidence of diarrhea without inducing constipation. However, a subsequent randomized, open-label, multicenter phase II trial revealed that prophylactic administration of acetorphan at 300 mg/day did not exert a beneficial effect on CID [64].

Budesonide, typically utilized in the treatment of inflammatory bowel diseases, has been explored for its potential role in CID management. In a phase I trial, fourteen patients experiencing stage 4 diarrhea due to CPT-11 were administered budesonide (9 mg oral dose each morning). Budesonide was observed to reduce the severity of diarrhea by at least two levels in 86% of the CPT-11-treated patients [65]. However, in a phase III randomized, double-blind, placebo-controlled trial involving patients with advanced colorectal cancer (CRC), oral budesonide did not manifest significant advantages in the prevention of CID, although the findings were somewhat encouraging [66].

#### 3.1.2. Intestinal Alkalization

CPT-11, its active metabolite SN-38, and the glucuronide conjugate SN-38G all possess an unstable α-hydroxy-δ-lactone ring, the integrity of which is susceptible to pH-dependent hydrolysis. The detrimental impact of SN-38 on intestinal epithelial cells is postulated to be the primary cause of delayed diarrhea associated with CPT-11 use. At physiological pH levels or higher, the less harmful carboxylic acid form of the compound predominates, whereas in an acidic environment, the more toxic lactone form is favored [67]. Consequently, modulating the equilibrium between these carboxylate isomers can attenuate the toxicity of CPT-11. Given that the intestinal absorption rates of CPT-11 and SN-38 are pH-sensitive, with absorption decreasing by over 65% when the pH exceeds 6.8, an alkaline intestinal milieu can effectively diminish the absorption of CPT-11 and SN-38 by the intestinal cells.

A case-control study involving lung cancer patients treated with CPT-11 utilized an alkalinization regimen consisting of sodium bicarbonate, magnesium oxide, water, and ursodeoxycholic acid administered for four days following CPT-11 treatment. This approach, which included intestinal alkalinization and controlled defecation, was found to markedly enhance the management of delayed diarrhea, as well as other side effects such as nausea, vomiting, and neutropenia [68]. Subsequent research involving patients with advanced gastrointestinal cancer who were on a CPT-11 treatment regimen revealed that the daily intake of 2 g of primary sodium bicarbonate powder, diluted in 250 mL of water for four days post-CPT-11 injection, resulted in a reduced incidence of grade 3 to 4 diarrhea to 16%, as opposed to the anticipated rate of 24% in large clinical phase III trials [69]. Nonetheless, a case report highlighted that intestinal alkalinization could significantly lower plasma levels of SN-38 and CPT-11 [70]. Moreover, the preventive regimen, while potentially effective, is notably cumbersome, necessitating the consumption of over 2–3 L of highly alkalized water daily throughout the treatment period [68,71].

#### 3.1.3. Transporter Inhibition

As previously discussed, the pharmacokinetics of CPT-11 are influenced by a multitude of transporters, including ABCB1, ABCC1, ABCC2, and ABCG2 [18]. The modulation of these transporters through inhibition by various compounds can significantly impact the pharmacokinetic profile of CPT-11, the biliary concentration of SN-38, and consequently, the risk of intestinal toxicity.

The diarrhea induced by CPT-11 is predominantly attributed to the direct cytotoxic effects of SN-38 on the intestinal tract. SN-38 and its glucuronide conjugate, SN-38G, are transported from systemic circulation into the biliary tract via ABCB1 and ABCC2, subsequently entering the intrahepatic circulation [18]. Probenecid, an established inhibitor of ABCB1 and ABCC2, has demonstrated potential in reducing biliary excretion of CPT-11 metabolites in preclinical studies [72,73]. In a phase I clinical study involving 37 CPT-11-treated patients who received intravenous cyclosporine (5 mg/kg over 3 days), only one case of grade 3 diarrhea was reported. The subsequent phase of the study proposed a dosing regimen of 100 mg/m^2^ of CPT-11 every two weeks [49]. When 34 patients were treated at this recommended dose, the incidence of grade 4 diarrhea was observed to be 3%. However, the study also noted that some patients had to discontinue participation due to cyclosporine-induced toxicity [74].

The transporter MRP2/ABCC2 plays a pivotal role in the biliary excretion of SN-38 and SN-38G, whereas its involvement in CPT-11 transport is considerably less [47]. Probenecid, an MRP2 inhibitor, when co-administered with CPT-11, has been shown to decrease the biliary excretion of CPT-11, SN-38, and SN-38G, thereby increasing their systemic concentrations. A reduction in the dosage of CPT-11 resulted in decreased intestinal SN-38 levels and a reduced incidence of CID, without significantly altering plasma SN-38 levels or the occurrence of myelosuppression compared to the control group [75].

#### 3.1.4. Enzyme Induction and Inhibition

##### β-Glucuronidase Inhibition

The active metabolite SN-38, derived from CPT-11, can be metabolized into its inactive glucuronide form, SN-38G, by the action of hepatic uridine diphosphate glucosyltransferase [34,76]. A body of research has shed light on the pivotal role of bacterial β-glucuronidase in the pathogenesis of delayed diarrhea. The intestinal microbiota has the enzymatic capability to convert SN-38G back into the active and potentially damaging SN-38, a process that can result in significant intestinal mucosal injury [77].

D-saccharic acid 1,4-lactone (SAL), recognized for its potential as a therapeutic agent for cerebral ischemia–reperfusion injury in rats, has also been investigated for its capacity to inhibit β-glucuronidase. Studies have shown that the concurrent use of SAL can markedly decrease the intestinal mucosal damage induced by CPT-11 in preclinical rat models [78].

There is an ongoing effort to develop targeted methods that selectively inhibit bacterial β-glucuronidase without adversely affecting the intestinal symbiotes or the activity of mammalian β-glucuronidase. Rasmussen et al. [79] synthesized a novel compound, uronic acid-noan glucoside, which has demonstrated a competitive inhibitory effect against Escherichia coli β-glucuronidase. However, the compound exhibited minimal inhibitory activity against mammalian β-glucuronidase derived from bovine liver. More recently, nicotinamide, isoniazid, and amoxapine have been identified as inhibitors of bacterial β-glucuronidase, with negligible effects on the mammalian counterpart. Despite these promising findings, the clinical efficacy and safety of these compounds await further evaluation in human trials [80].

##### UGT1A1 Induction

As previously mentioned, UGT1A1 plays a crucial role in the metabolic pathway of CPT-11, facilitating the conversion of the cytotoxic metabolite SN-38 into a less harmful glucuronide form, SN-38G. Chrysin, despite its low oral bioavailability [81], has demonstrated the ability to upregulate UGT1A1 activity [82]. Consequently, chrysin can selectively enhance the glucuronidation of SN-38 to SN-38G within the gastrointestinal tract through the induction of UGT1A1. This process may mitigate intestinal mucosal damage and the onset of delayed diarrhea without impacting the systemic levels of SN-38 or its concentration within tumors. Other agents known to induce UGT1A1, including phenobarbital—as previously discussed—and glucocorticoids, have been the subject of clinical investigations. Notably, dexamethasone has not been found to significantly alter the area under the curve (AUC) for SN-38 and CPT-11, suggesting minimal influence on the enzymes CYP3A4 and UGT1A1 through which these substances are metabolized [83,84].

##### Carboxylesterases Inhibition and Activation

Intestinal carboxylesterase (hiCE) is instrumental in the conversion of CPT-11 to its active metabolite, SN-38, within the intestinal lumen. Evidence from human intestinal biopsies has confirmed the presence of hiCE, and in vitro assays have substantiated the direct conversion of CPT-11 to SN-38 by this enzyme. It is hypothesized that the inhibition of hiCE, thereby diminishing the levels of the active metabolite SN-38, could potentially lower the incidence of delayed diarrhea [85].

In this regard, Wadkins et al. [86] have synthesized a suite of seven hiCE inhibitors, all sulfonamide-based, which exhibit over 200-fold selectivity for hiCE relative to hepatic carboxylesterase. These inhibitors do not adversely affect human acetylcholinesterase or butyrylcholinesterase activities. Among the lead compounds developed, four nitrophenol derivatives have shown particular promise, with compound 3 demonstrating superior inhibitory efficacy against hiCE compared to rabbit liver carboxylesterase, leaving 14% of CES activity intact, as reported by Yoon et al. [87]. More recently, a new class of fluorene analogues has been introduced as hiCE inhibitors, showcasing enhanced potency and efficiency [85]. The efficacy and safety profiles of these novel inhibitors in animal models and, ultimately, in human studies remain to be established.

##### CYP3A4 Inducers

Anti-epileptic medications, such as phenytoin, carbamazepine, and phenobarbital, which are known to induce the activity of the cytochrome P450 3A4 (CYP3A4) enzyme, have been utilized in combination with or without dexamethasone. This treatment approach has been observed to elevate the recommended dosage of CPT-11 for patients who have not previously received these anti-epileptic drugs. Specifically, the dose of CPT-11 can be escalated from 350 mg/m^2^ to 750 mg/m^2^ when administered every three weeks, due to the increased metabolic clearance induced by these CYP3A4 inducers [83]. This induction effect results in an accelerated elimination of CPT-11 and a subsequent reduction in the area under the curve (AUC) for its active metabolite, SN-38.

In a separate study, a parallel increase in the clearance rate of CPT-11 and the dosage of temozolomide was noted. The escalated dosage regimen of temozolomide was 500 mg/m^2^, administered every fifteen days over a 28-day cycle, in patients undergoing treatment with enzyme-inducing anti-epileptic drugs [88]. These findings underscore the importance of considering the impact of concomitant medications on the pharmacokinetics and optimal dosing strategies of chemotherapeutic agents.

##### COX-2 Inhibition

Cyclooxygenase-2 (COX-2) is notably overexpressed in metastatic colorectal cancer, where it contributes to tumorigenesis through multiple pathways, notably the synthesis of prostaglandins E2 (PGE2) and thromboxane A2 (TXA2) [89]. The administration of CPT-11 has been linked to upregulation of COX-2 in intestinal epithelial cells, which leads to elevated PGE2 levels. This increase in PGE2 induces heightened chloride secretion and diminished sodium absorption in the intestinal cells, a sequence of events that culminates in diarrhea. Research by Trifan et al. [90] has shown that celecoxib, a COX-2 inhibitor, can curtail the production of prostaglandins within the intestinal mucosa. It also lessens the colon’s inflammatory response and markedly reduces the frequency of diarrhea in mice, with the severity of diarrhea correlated to the administered dose of celecoxib.

Further investigation in Ward CRC rat models demonstrated that oral celecoxib, administered at a dosage of 30 mg/kg daily in two divided doses, could lessen the toxicity associated with CPT-11 and bolster antitumor efficacy, alongside improved survival rates at doses that would otherwise be lethal [91]. However, subsequent clinical studies did not replicate these beneficial effects, failing to demonstrate any significant amelioration in the management of CID or in the treatment of colorectal cancer [92]. Diarrhea persisted as the primary non-hematological side effect. The lack of therapeutic improvement may stem from the possibility that COX-2 inhibition alone is insufficient to mitigate CID, or potentially due to inadequate delivery of celecoxib to achieve the desired protective effect on the target tissue [92].

#### 3.1.5. Alteration of Intestinal Microflora

##### Prebiotics and Antibiotics

A study has demonstrated the potential benefits of Lactobacillus casei strain Shirota (LcS) in a rat model. The rats were administered LcS intragastrically at a dosage of 1.64 × 10^11^ colony-forming units (CFU) per 0.5 g in 3 mL saline for a period of 28 days. Commencing from the 14th day, the rats were also given CPT-11 at a dosage of 100 mg/kg for four consecutive days. A control group was simultaneously treated with CPT-11 and an equivalent volume of saline. The findings indicated that LcS treatment led to a significant reduction in weight loss, and the rats exhibited a markedly higher food intake compared to the control group. Moreover, the LcS group showed an improvement in the symptoms of delayed diarrhea associated with CPT-11, a beneficial effect that may be attributed to the inhibition of β-glucuronidase activity by LcS, an enzyme produced by the intestinal flora [93].

In another investigation, VSL#3, a commercially available probiotic formulation comprising Lactobacillus, Bifidobacterium, and Streptococcus species, was assessed for its effects on CPT-11-treated rats. The study found that VSL#3 could enhance glandular proliferation, mitigate weight loss, and reduce the severity of diarrhea, intestinal cell apoptosis, mucin secretion, and the increase of goblet cells in the jejunal crypts induced by CPT-11. However, the protective effects of VSL#3 were most pronounced when the probiotic was administered both prior to and following the chemotherapy regimen [94].

##### Antibiotics

The clinical use of antibiotics serves a dual purpose in the management of CID by targeting microorganisms that produce β-glucuronidase, an enzyme that can convert the inactive metabolite SN-38G back to its active form, SN-38, thereby exacerbating diarrhea. By reducing the intestinal bacterial load and β-glucuronidase activity, the concentration of SN-38 in the gut can be diminished, which may alleviate diarrheal symptoms [95]. In a clinical study involving patients with colorectal and small cell lung cancer treated with CPT-11, the administration of neomycin (500 mg, twice daily) in subsequent treatment cycles effectively prevented the recurrence of severe diarrhea [96].

In vivo research has shown that amoxicillin can mitigate the toxicity of CPT-11 and enhance its antitumor efficacy in tumor-bearing mice to a certain degree. This effect is associated with the inhibition of β-glucuronidase activity in various bacterial strains, including Escherichia coli, enterococci, streptococci, and staphylococci [97]. However, the alleviation of CID by antibiotics is not solely attributable to the inhibition of microbial β-glucuronidase. An animal study utilizing three different diarrhea models demonstrated that while streptomycin could ameliorate intestinal toxicity, it did not inhibit β-glucuronidase activity [98]. This discrepancy may stem from the significant variability in the catalytic efficiency, substrate binding, and reaction rates of β-glucuronidase among different bacterial species [99]. Other antibiotics, such as levofloxacin [100] and cefaclor [101], have also been utilized clinically to treat CID.

#### 3.1.6. Prevention of Direct Intestinal Exposure

Activated charcoal (AC) is a widely utilized oral adsorbent with the capacity to adsorb the active metabolite SN-38 within the intestine, thereby mitigating intestinal mucosal injury and reducing the incidence of CID. AC also augments the clearance of SN-38 by engaging with intestinal capillaries and impeding the enterohepatic recirculation of the drug [102]. In a pediatric study involving patients undergoing CPT-11 chemotherapy, those in the intervention group were administered 250 mg of AC three times daily concurrently with CPT-11. This regimen resulted in a significant reduction of grade 3 and 4 diarrhea cases to 4.4% within the AC group, as opposed to the control group. Additionally, the intervention group experienced fewer chemotherapy discontinuations (6.6% vs. 52.3%). Consequently, AC has been shown to improve CPT-11 compliance and diminish both the frequency and severity of CID. Nonetheless, AC’s efficacy is not absolute, it can also adsorb other concurrently administered oral medications, and the requirement for thrice-daily administration can be burdensome [103].

AST-120 (clemizine) is an alternative oral carbonaceous adsorbent that has been investigated for its potential to prevent delayed diarrhea. Clemizine has demonstrated a significant adsorption capacity for CPT-11 both in vitro and within the rat gastrointestinal tract. In rats administered with clemizine, there was approximately a 50% reduction in the frequency of diarrhea compared to untreated controls, with a corresponding decrease in severity [104]. In a clinical trial, the administration of two grams of clemizine, given in three divided doses daily, was found to decrease the occurrence of CID during and post-CPT-11 treatment, with minimal impact on the pharmacokinetics of CPT-11 and its metabolites [105].

#### 3.1.7. Cytokine and Growth Factors Induction and Inhibition

Thalidomide, a synthetic derivative of glutamic acid, has been recognized for its potential to mitigate the intestinal pathological changes induced by CPT-11. It exerts its effects by inhibiting the production of inflammatory cytokines within the intestine, reducing apoptosis of intestinal epithelial cells, and modulating immune and angiogenic responses [106]. Preliminary clinical studies have affirmed thalidomide’s utility beyond its antitumor properties, highlighting its anti-angiogenic and immunomodulatory capabilities, and its capacity to alleviate CID [107]. While one study has reported a significant reduction in the metabolic conversion of CPT-11 to SN-38 with thalidomide treatment, others have not observed a substantial impact on the pharmacokinetics of CPT-11 [108,109].

In an additional study, Velafermin, a fibroblast growth factor-20, was administered at a dosage of 16 mg/kg prior to CPT-11 treatment. This approach was found to ameliorate gastrointestinal mucositis, diarrhea, and mortality associated with CPT-11 in tumor-bearing DA rats. Although rats treated with verapamil experienced severe or moderate diarrhea, the onset was delayed, and the condition was less severe and shorter in duration. Notably, verapamil did not influence tumor proliferation. Various doses of verapamil were assessed, with some showing diminished effectiveness in reducing the severity and mortality of CID. Interestingly, certain dosages were associated with increased diarrhea and mortality [110].

Interleukin-15 (IL-15) is a cytokine that has demonstrated a pronounced protective effect against CPT-11-induced intestinal toxicity and has the potential to moderately boost the antitumor efficacy in advanced colorectal cancer models [111]. JBT3002, a novel synthetic bacterial lipopeptide, has been shown to stimulate the production of IL-15. This compound can protect the integrity of the intestinal epithelium, prevent damage to the intestinal epithelium and mucosal lamina propria, and is tolerable at higher intravenous doses [112].

#### 3.1.8. Other Chemical Drug Treatment Options

In the realm of preclinical research, the supplementation of diets with low concentrations of fish oil—specifically at levels of 3% or 6%—has been observed to enhance the regression of MCF7 human breast cancer xenografts in nude mice, both prior to and during the administration of CPT-11. This augmentation of CPT-11’s therapeutic efficacy is accompanied by a reduction in the pathological damage to the intestinal tissue [113].

L-Glutamine is categorized as a conditionally essential amino acid, particularly in conditions of stress where the body’s synthesis may not meet the increased demands [114]. It is an indispensable nutrient that supports the growth, differentiation, and the maintenance of integrity and barrier function of the intestinal mucosal epithelium. Studies have demonstrated its role in facilitating electrolyte absorption in animals subjected to experimentally induced diarrhea [94].

Phloroglucinol, a myophilic smooth muscle antispasmodic agent, has been identified in a study to ameliorate diarrhea and normalize electrolyte levels [115]. This drug does not exert anticholinergic effects and is less likely to cause adverse cardiovascular effects such as tachycardia, hypotension, or arrhythmia. Given these attributes, phloroglucinol is deemed more appropriate for use in elderly patients with pre-existing cardiovascular or cerebrovascular conditions [116]. Figure 3 illustrates the major chemotherapeutic agents for the treatment of CPT-11-induced diarrhea. (A summary of the mechanisms of action and limitations of major chemotherapeutic agents for the treatment of CPT-11-induced diarrhea can be found in Appendix A).

### 3.2. Traditional Chinese Medicine Treatment of Delayed Diarrhea

Despite the availability of numerous chemical drugs for the prevention and treatment of CID, there remains no universally recognized and consistently effective standard treatment. The efficacy of individual drugs in isolation is challenging to ascertain. However, modern pharmacological studies have established that traditional Chinese medicine (TCM) possesses a unique multi-target and multi-pathway approach. TCM is tailored to individual patient profiles, which often results in favorable outcomes in the treatment of diarrhea caused by CPT-11. Figure 4 outlines the major traditional Chinese medicine treatment of delayed diarrhea.

#### 3.2.1. Single Component and Active Components of Traditional Chinese Medicine

##### *Hypericum perforatum* L. (St. John’s Wort) Extract

*Hypericum perforatum* L., commonly referred to as St. John’s wort, is a member of the Guttiferae family and has been traditionally used for its diarrhea-treating properties [117]. Experimental rat studies have substantiated the efficacy of St. John’s wort in this context. In these experiments, a control group was treated solely with CPT-11, whereas a treatment group additionally received oral St. John’s wort. The results indicated a significant reduction in the expression of the inflammatory marker TNF-α mRNA in rats administered with St. John’s wort. Furthermore, it was observed to partially inhibit apoptosis in intestinal epithelial cells, thereby mitigating the intestinal damage caused by CPT-11 [118].

##### Berberine

Berberine, a principal active constituent of the traditional Chinese medicinal herb *Coptis chinensis* (huanglian), has been ascribed with a spectrum of pharmacological activities, including anti-inflammatory, antioxidant, anticancer properties, lipid metabolism regulation, and energy balance maintenance [119]. In an animal model, berberine was found to mitigate the mucosal structural damage, ulceration, and neutrophil infiltration induced by CPT-11. It also enhances mucosal barrier function by increasing the number of goblet cells, preserving transepithelial electrical resistance (TEER), reducing permeability, and upregulating tight junction proteins. Berberine was noted to reduce fecal SN-38 levels, which may be linked to a decrease in the activity of β-glucuronidase and the corresponding bacterial population. Notably, berberine preserves the anticancer efficacy of CPT-11 while simultaneously reducing its intestinal toxicity in xenograft tumor models [120].

##### Curcumin

Curcumin, a polyphenolic compound derived from the rhizome of *Curcuma longa* L. (turmeric), is renowned for its diverse pharmacological effects, including anti-inflammatory, antioxidant, and antitumor activities. In the establishment of a mouse model for delayed diarrhea induced by CPT-11, a group treated with curcumin prophylactically showed protective effects against the symptoms and pathophysiology of the condition [121]. Further animal studies demonstrated that curcumin could effectively alleviate CID symptoms and intestinal mucosal structural aberrations in nude mice. Curcumin was found to upregulate the expression of P4HB and PRDX4 in the small intestine, enhance cell morphology, inhibit apoptosis, maintain mitochondrial membrane potential, and reduce the rise of reactive oxygen species (ROS) levels provoked by CPT-11 (20 μg/mL) in vitro. Additionally, curcumin increases the expression of molecular chaperone proteins such as GRP78, P4HB, and PrDX4, and suppresses the expression of apoptosis-related proteins like CHOP and cleaved caspase-3, thereby interrupting the NF-κB signaling pathway and safeguarding cells against CPT-11-induced apoptosis [122].

##### Hesperidins

Hesperidin, the predominant flavonoid in *Citrus reticulata Blanco* (chenpi), is recognized for its vascular protective properties, enhanced lymphatic circulation, and demonstrated anti-inflammatory and antiviral activities. It is postulated that CPT-11’s dose-restricted diarrhea is associated with the exposure of the active metabolite SN-38 to the intestinal tract. Hesperidin has been shown to modulate the biliary excretion transporters of CPT-11 and its metabolites, thereby influencing the pharmacokinetics of both CPT-11 and SN-38 [123,124].

#### 3.2.2. Compound Traditional Chinese Medicine and Extract

##### Huangqin Decoction

Huangqin Decoction (HQD) is a complex traditional Chinese medicinal formula consisting of *Scutellaria baicalensis* Georgi, *Glycyrrhiza uralensis* Fisch, *Paeonia lactiflora* Pall, and *Ziziphus jujube* Mill in a dry weight ratio of 3:2:2:2. With a history spanning over 1800 years, HQD has been widely utilized in China for the treatment of gastrointestinal disorders characterized by diarrhea, nausea, abdominal cramps, and vomiting [125]. In a study involving ICR rats with CID, HQD was administered one day prior to the experiment and continued for eight days. The findings indicated that HQD-treated rats exhibited a reduced rate of body weight loss and intestinal mucosal injury. There was a significant increase in the expression of nitric oxide (NO) in the colon, a decrease in proliferating cell nuclear antigen (PCNA) expression, and an elevated count of blood neutrophils [126]. In another animal experiment, the concurrent administration of HQD (10 g/kg, twice daily) at the onset of CPT-11 significantly mitigated delayed diarrhea in rats, although it was ineffective in preventing acute diarrhea during the initial two days. Researchers employed GC/MS and LC/MS metabolomics to analyze serum metabolite changes in male SD rats before and after HQD treatment, suggesting that HQD could mediate metabolic alterations by normalizing amino acid, lipid, and bile acid metabolic pathways [127]. PHY906, a derivative of HQD, has demonstrated its utility as a modulator of chemotherapeutic agents, particularly in alleviating cancer therapy-induced nausea, vomiting, and diarrhea [128,129]. In a mouse model of allogeneic colon transplantation, oral PHY906 (dosed at 50, 500, or 1000 mg/kg, twice daily) was shown to attenuate CPT-11-induced gastrointestinal toxicity through multiple mechanisms, including the inhibition of NF-κB, COX-2, and iNOS inflammatory pathways, and the promotion of intestinal progenitor cell regeneration via the upregulation of Wnt signaling components, with a particular emphasis on Wnt3a [130]. A phase I, multicenter, double-blind, randomized, placebo-controlled crossover study involving 17 patients with advanced colorectal cancer treated with PHY906 in combination with CPT-11 and 5-FU/IFL regimens showed a reduction in the overall incidence of grade 3 or 4 diarrhea and decreased reliance on antidiarrheal medications such as loperamide and lomotil. Notably, PHY906 did not alter the pharmacokinetics of CPT-11 or its metabolite SN-38. However, given the small patient sample size in this trial, further large-scale randomized trials are necessary to fully assess the benefits of PHY906 in the context of CID [131].

##### Shengjiang Xiexin Decoction

Shengjiang Xiexin Decoction (SXD), a classic compound from the traditional text Shang Han Lun, is composed of eight Chinese herbal medicines and is extensively used in contemporary clinical practice for the treatment of gastroenteritis, ulcerative colitis, and diarrhea [132]. In a rat model of diarrhea, SXD, administered at dosages of 5, 10, or 15 g/kg per day, was found to promote intestinal cell proliferation while inhibiting intestinal cell apoptosis and β-glucuronidase activity, thereby preventing delayed diarrhea induced by CPT-11 in a dose-dependent manner [133]. In an experiment utilizing a CT26 colon cancer mouse model, mice in the experimental group were given SXD (10 g/kg, twice daily) three days prior to CPT-11 administration for a total of eight days. In comparison to the model and control groups, which received equivalent volumes of normal saline, CPT-11 injection led to significant diarrhea in the model group. The SXD group exhibited significantly lower diarrhea scores, less severe intestinal mucosal damage under light microscopy, decreased levels of TNF-α, and increased levels of IL-10, effectively alleviating neutropenia [134]. In a randomized controlled trial involving 115 patients treated with CPT-11 combined with 5-fluorouracil and calcium 1-folinate, SXD (100 mL, twice daily) was shown to significantly reduce the incidence of delayed diarrhea in patients with UGT1A1*28 or UGT1A1*6 mutations, without compromising the clinical response to chemotherapy [135].

##### Banxia Xiexin Decoction

Banxia Xiexin Decoction (BXD), a formulation comprising seven medicinal ingredients, is a potent compound prescription in traditional Chinese medicine used to address conditions such as gastroenteritis, ulcerative colitis, vomiting, and diarrhea [136]. In experiments involving mice with small-cell lung cancer (SCLC) induced to experience diarrhea by CPT-11, BXD was observed to ameliorate the condition. This improvement is hypothesized to be due to the inhibition of COX-2 expression in the colonic tissue and a reduction in the local concentration of SN-38 [137]. A clinical study involving 27 patients with recurrent SCLC undergoing CPT-11 chemotherapy reported that BXD, administered prior to the second chemotherapy cycle, effectively prevented and treated delayed diarrhea caused by CPT-11. Out of six patients who developed delayed diarrhea, four were relieved after BXD treatment. However, the small sample size in this study necessitates further evaluation in larger, high-quality, randomized controlled trials to confirm the efficacy of BXD in managing CPT-11-induced delayed diarrhea [138].

##### Gegen Qinlian Decoction

Gegen Qinlian Decoction (GQD) is a traditional Chinese medicinal formula that *includes Pueraria* lobata (Gegen), *Scutellaria baicalensis* (Huangqin), *Coptis chinensis* (Huanglian), and *Glycyrrhiza uralensis* (Gancao). Originating from the Shan Han Lun, a text dating back to the Han Dynasty (202–220 BC), GQD is widely utilized for gastrointestinal disorders, particularly diarrhea [139]. In a mouse model of CID, GQD extract administration for five days led to a significant decrease in the levels of inflammatory cytokines such as IL-1β, COX-2, ICAM-1, and tumor necrosis factor-α within the intestinal tissue. Additionally, GQD demonstrated antioxidant properties, activated the Keap1/Nrf2 pathway, and enhanced the intestinal barrier function by upregulating the expression of tight junction proteins like ZO-1, HO-1, and occludin. The GQD extract also exhibited a potent inhibitory effect on hCE2 in vitro, with an IC50 value of 0.187 mg/mL, suggesting its potential in mitigating hCE2-mediated severe diarrhea [140]. Moreover, the extract of GQD has been shown to synergistically inhibit the growth of colon cancer when used in conjunction with CPT-11. In vitro studies have also highlighted GQD’s significant inhibitory effect on CES2 [141].

##### Xiao Chaihu Decoction

Xiao Chaihu Decoction (XCD), a blend of seven Chinese medicinal herbs, is the principal prescription for Shaoyang disease as described in the Shang Han Lun of the Han Dynasty [142]. Extensive clinical and experimental studies have validated XCD’s efficacy in treating liver and digestive system diseases. In an animal study, the experimental group received XCD at a dosage of 1500 mg/kg (based on crude drug) once daily for 17 days, while other groups, except for the normal control, were injected with CPT-11 to induce delayed diarrhea from the 4th to the 10th day. The results indicated that XCD significantly reduced the rate of hematochezia and improved intestinal mucosal injury in the treated mice [143].

##### Other TCM Preparations for Treatment

Sishen Pill, a classic formula in traditional Chinese medicine, is recognized for its effectiveness in treating diarrhea and has shown promise in managing irritable bowel syndrome and ulcerative colitis. In an animal model using ICR mice, Sishen Pill demonstrated preventive and therapeutic effects on CPT-11-induced delayed diarrhea, potentially through reducing intestinal β-glucuronidase activity and the levels of IL-1β and TNF-α [144].

Jiawei Xianglian Decoction (JWXLD), a combination of six medicinal ingredients, is a clinically used drug in China effective against diarrhea. Studies have found that JWXLD at dosages of 0.12, 0.23, and 0.46 g significantly mitigated the severity of CID and altered the levels of Lactobacillus and Bifidobacterium in mice, effects that were reversible with JWXLD. Furthermore, JWXLD was shown to reduce β-glucuronidase activity. Histopathological assessments revealed that JWXLD could significantly lessen the severity of intestinal mucosal injury caused by CPT-11 in rats [145]. Senfu Zhulin San [146] and Renshen Jianpi Pill [147] are also utilized in Chinese clinical practice to treat CID, likely through mechanisms involving the regulation of intestinal flora and the inhibition of inflammation.

Hange-shashin-to, a formula consisting of seven Chinese herbal medicines, is used in Japan for treating diarrhea and acute gastroenteritis. In animal studies, Hange-shashin-to (1 g/kg, twice daily) showed protective effects against CPT-11-induced intestinal toxicity by inhibiting β-glucuronidase activity, leading to reduced weight loss, improved anorexia, and delayed onset of diarrhea symptoms [148]. Sairei-to, a preparation of 12 traditional Chinese herbs, is used in Japan for severe diarrhea and various inflammatory conditions, including rheumatoid arthritis, systemic lupus erythematosus, and nephrotic syndrome [149]. A preclinical study in male Wistar rats indicated that Sairei-to could alleviate CPT-11-induced delayed diarrhea, possibly through the inhibition of bacterial β-glucuronidase [150]. (A summary of the proved effects of single Chinese herbs and traditional Chinese medicine compound prescriptions on CPT-11-induced diarrhea can be found in Appendix A).

### 3.3. Structural/Chemical Modification and Novel Drug Delivery Methods

To address the challenges of low bioavailability and enterotoxicity associated with the anticancer drug CPT-11 and its active metabolite SN-38, various novel dosage forms and drug delivery systems have been explored. These innovations include the development of liposomal formulations, polymer conjugates, nanoparticles, dendrimers, and the utilization of peptides and carbohydrates. Figure 5 summarizes the structural modifications and novel drug delivery methods explored to enhance the efficacy and reduce the toxicity of irinotecan and its metabolites.

#### 3.3.1. Novel Drug Delivery of CPT-11

##### Liposomal Formulations of CPT-11

Irinotecan liposomes, such as Nal-IRI (ONIVYDE), were approved in 2015 [151] and have since been employed as a second-line treatment for metastatic pancreatic cancer [152,153]. New liposomal preparations of CPT-11, like CPX-1, have demonstrated antitumor activity in phase I studies by combining CPT-11 with fluorouracil, improving outcomes in patients with advanced solid tumors [154]. Additionally, highly stable nanoparticles/liposomes integrated with convection-enhanced delivery (CED) have been recognized for their potential to prolong drug retention in tissues and reduce toxicity [155]. These liposomal formulations are designed to enhance target specificity, thereby minimizing systemic toxicity. Despite the potential shown in vitro, many of these carriers have yet to fulfill their promise in vivo.

##### Nanoparticles of CPT-11

Wang et al. have developed novel nanoparticles using hyaluronic acid, poly (lactic acid-glycolic acid), chitosan, and Pluronic F-127 as carriers for CPT-11 and doxorubicin. Hyaluronic acid chemical transport (HyACT^®^) has emerged as a carrier system for CPT-11, enhancing the reactivity with CD44-positive tumor cells and improving progression-free survival rates in metastatic colorectal cancer when used in combination therapies [156,157]. In a randomized phase II trial, hyaluronic acid-modified CPT-11 was found to enhance progression-free survival in patients with metastatic colorectal cancer resistant to 5-fluorouracil [158].

##### Polymer Conjugates of CPT-11

The covalent attachment of polyethylene glycol (PEG) molecules has been utilized to enhance the systemic circulation and reduce clearance of CPT-11. PEGylated liposomal CPT-11 preparations, such as MM-398, have shown improved cytotoxicity in mouse brain metastasis models compared to CPT-11 monotherapy [159,160]. In a phase I clinical and pharmacokinetic study, IHL-305, a novel PEGylated liposome formulation of CPT-11, demonstrated a safe repeat dosing regimen over 4 or 2 weeks [161]. Zhang et al. have explored the combination of CPT-11 with fatty acids to increase lipophilicity and facilitate self-assembly in an aqueous environment, protecting CPT-11 from carboxylesterase-mediated hydrolysis and enhancing intracellular accumulation and cytotoxicity [162]. Zashikhina et al. have developed self-assembled poly (l-lysine)-b-poly (l-leucine) (plys-b-pleu) polymers that exhibit no cytotoxicity in tested cell lines and maintain in vitro antitumor activity similar to that of free CPT-11 [163].

Strategies to enhance CPT-11 activity, such as adenovirus-mediated β-glucuronidase expression in tumors [164] and gene-directed enzyme/prodrug therapy (CES/CPT-11) [165], may indirectly improve antitumor efficacy and ameliorate CID. The hypothesis is that targeted administration and the use of liposomes could reduce the concentration of CPT-11 and SN-38 in bile, thereby minimizing intestinal damage.

#### 3.3.2. Novel Drug Delivery of SN-38

##### Liposomal Formulations of SN-38

Machmudi et al. have constructed polyethylene glycol polyamide (PAMAM) dendrimers conjugated with SN-38 and targeting moieties BR2 and CyLoP1. These dendrimers have shown enhanced cytotoxicity and cellular uptake in mouse colon cancer (CT26) cell lines compared to native SN-38, with in vivo studies demonstrating improved drug accumulation at the tumor site and increased antitumor efficacy [143]. Zhang et al. have developed a liposome-based SN-38 preparation (LE-SN-38) that has exhibited superior efficacy in enhancing cytotoxicity against various tumor cell lines and in treating xenogeneic mice models [166]. Sun et al. have created amorphous solid disodium glycyrrhizinate and SN-38 self-assembled micelles (Na2GA/SN-38-BM) with favorable pharmacokinetics and distribution properties, demonstrating enhanced cytotoxicity against tumor cells and significant tumor growth inhibition [167].

##### Nanoparticles of SN-38

Karki et al. have investigated the loading of SN-38 onto graphene oxide (GOS) modified with polyvinylpyrrolidone (PVP) or β-cyclodextrin (β-CD). Their studies revealed that SN-38 loaded onto GO-PVP nanocarriers exhibited higher cytotoxic activity against human breast cancer cells (MCF-7) than GO-β-CD nanocarriers, suggesting the potential of GO-PVP as an effective drug delivery system [168]. Naumann et al. have conjugated SN-38 to gold nanoparticles using oligonucleotides specific to Ewing’s sarcoma cells, allowing for the targeted release of SN-38 and its subsequent inhibition of topoisomerase, with effective and selective drug release observed in both in vitro and in vivo conditions [169]. Furthermore, coupling was performed with monoclonal antibodies (Labetuzumab-SN-38 immunoconjugates [170]), along with a variety of other preparations and methods [171] (A summary of the mechanisms of action and effects on diarrhea relief of new drug delivery systems for CPT-11 and SN-38 can be found in Appendix A).

#### 3.3.3. Impact on Diarrhea Risk

Despite the promising strategies offered by advanced drug delivery systems for enhancing the therapeutic efficacy of CPT-11 and SN-38 and reducing systemic toxicity, their impact on diarrhea risk remains a significant consideration. Systems such as liposomal formulations and nanoparticles, which reduce systemic exposure to the active metabolite SN-38, may potentially lower the incidence of CID. However, clinical study results have been inconsistent, with some trials reporting reduced severe diarrhea and others showing no significant difference compared to conventional formulations [151]. Moreover, the long-term clinical outcomes and specific impact on diarrhea incidence require further elucidation.

In summary, while novel drug delivery systems show promise in enhancing the therapeutic efficacy of CPT-11 and SN-38 and reducing systemic toxicity, their impact on diarrhea risk remains a critical factor. Further research is needed to fully understand the long-term effects of these delivery systems on CID risk and to optimize their clinical application [154,158,172].

## 4. Discussion

### 4.1. Clinical and Preclinical Strategies for Blocking or Treating

The management of irinotecan-induced diarrhea (CID) encompasses a diverse array of therapeutic approaches, each with varying levels of clinical validation and developmental maturity. Clinically validated treatments, such as loperamide and octreotide, have demonstrated robust efficacy in managing acute symptoms and are widely adopted in clinical practice. These treatments have undergone rigorous clinical trials, ensuring their safety and effectiveness are well-established [11,12]. For instance, loperamide, a synthetic opioid, effectively reduces intestinal motility and promotes fluid reabsorption, thereby alleviating acute diarrhea episodes. Similarly, octreotide, a somatostatin analog, mitigates gastrointestinal secretion and motility, offering significant relief in cases of severe diarrhea [13].

In contrast, several innovative strategies remain in the preclinical stage, showcasing promising potential but requiring further validation. Enzyme inhibitors targeting β-glucuronidase and UGT1A1 inducers have demonstrated substantial efficacy in reducing the severity and frequency of CID in preclinical models [31,128]. These approaches aim to modulate the metabolism of irinotecan’s active metabolite, SN-38, thereby mitigating its toxic effects on the intestinal mucosa. Additionally, advanced drug delivery systems, such as liposomal formulations and nanoparticles, are being explored to enhance the targeted delivery of irinotecan while minimizing systemic toxicity [47,156]. These systems hold the potential to reduce the incidence of CID by limiting the exposure of the gastrointestinal tract to SN-38.

Traditional Chinese medicine (TCM) also presents a unique avenue for CID management, with formulations like Huangqin Decoction and Shengjiang Xiexin Decoction demonstrating efficacy in both preclinical and clinical settings [129,173]. These herbal preparations leverage a multi-target approach, modulating inflammation, enhancing intestinal barrier function, and reducing mucosal injury. However, their long-term safety and efficacy profiles are still under evaluation, and standardization of these formulations remains an ongoing challenge [50].

Distinguishing between clinically validated and preclinical strategies is essential for guiding clinical practice and future research. While clinically validated treatments provide immediate relief and are supported by substantial evidence, preclinical strategies offer innovative solutions that could significantly improve patient outcomes once validated. Future research should focus on bridging the gap between these categories, ensuring that emerging therapies are rigorously tested and standardized before clinical application.

### 4.2. Long-Term Efficacy and Safety Considerations

The long-term efficacy and safety of therapeutic interventions for irinotecan-induced diarrhea (CID) are critical factors in optimizing patient outcomes and ensuring sustained therapeutic benefits. While many experimental treatments show promise in short-term studies, the availability of long-term data remains limited for several emerging strategies. For instance, conventional treatments such as loperamide and octreotide have demonstrated consistent efficacy in managing acute symptoms but may face challenges in maintaining long-term effectiveness due to potential tolerance development or side effects [11,12]. In contrast, novel approaches like enzyme inhibitors and advanced drug delivery systems, although highly effective in preclinical models, are still in the early stages of clinical evaluation, and their long-term impacts on patient health are yet to be fully elucidated [31,47].

Traditional Chinese medicine (TCM) formulations, which have garnered attention for their holistic and multi-target effects, also present unique considerations for long-term use. While TCM has shown potential in reducing the severity and frequency of CID in both preclinical and clinical settings [128,129], the long-term safety profile remains an area of active investigation. The complexity of herbal compositions and the potential for interactions with other medications necessitate careful monitoring and further research to establish their long-term efficacy and safety [173].

Moreover, the potential for delayed side effects cannot be overlooked. For example, prolonged use of certain medications may lead to gastrointestinal dysbiosis or other systemic complications, which could exacerbate the overall burden of treatment [50]. Therefore, long-term monitoring and follow-up studies are essential to identify and mitigate any adverse effects that may emerge over time.

In summary, while significant progress has been made in developing treatments for CID, the long-term efficacy and safety of these interventions remain areas of ongoing research. Future studies should prioritize the collection of long-term data to inform clinical practice and guide the development of sustainable, patient-centered treatment strategies. This approach will be crucial in addressing the multifaceted challenges of managing CID and improving the quality of life for patients undergoing irinotecan therapy.

### 4.3. Comparative Analysis of Therapeutic Approaches for Irinotecan-Induced Diarrhea

The management of irinotecan-induced diarrhea (CID) necessitates a thorough evaluation of the diverse therapeutic strategies available, each with distinct advantages and disadvantages. Conventional treatments, such as loperamide and octreotide, remain pivotal for their rapid alleviation of acute symptoms, supported by extensive clinical validation. However, their efficacy may wane over time due to potential drug tolerance and side effects, and they predominantly address symptoms rather than underlying mechanisms.

Traditional Chinese medicine (TCM) provides a holistic, multi-target approach, modulating inflammation and enhancing intestinal barrier function. While TCM formulations like Huangqin Decoction and Shengjiang Xiexin Decoction have shown efficacy in both preclinical and clinical settings, challenges remain in standardization and long-term safety assessment.

Novel drug delivery systems, including liposomal formulations and nanoparticles, offer enhanced targeted delivery and reduced systemic toxicity, potentially lowering the incidence and severity of CID. Despite promising preclinical results, clinical outcomes have been inconsistent, highlighting the need for further validation. (The advantages and disadvantages of different treatment methods can be found in Table 1).

Future research should focus on integrating these approaches to develop personalized treatment strategies that optimize patient outcomes. 

## 5. Conclusions

It is undeniable that nearly two decades after its introduction, CPT-11 remains a cornerstone in the arsenal of cytotoxic anticancer drugs, particularly for the treatment of advanced colon cancer and certain solid tumors [174]. The metabolic pathway of CPT-11 is intricate, involving numerous factors, leading to considerable inter-individual variability in pharmacokinetics and posing challenges for personalized therapeutic regimens [173]. Despite this, recent advancements have been made in tailoring treatment combinations to better suit diverse patient populations. The oral formulation of CPT-11 is also gaining traction. Yet, the drug is not without serious side effects, most notably neutropenia and diarrhea. CID is often severe, necessitating dose reductions, treatment omissions, and hospitalizations. These complications diminish the therapeutic efficacy of CPT-11, escalate healthcare costs, and degrade the quality of life for patients. Our understanding of the pathophysiology of CID has advanced, highlighting the interplay of inflammation and dysbiosis. Although numerous products are in development to treat or prevent CID, delayed diarrhea continues to be a significant challenge. Grade 3–4 diarrhea is frequently observed even with premedication, underscoring the limitations of current treatments. The complex metabolism of CPT-11, involving various enzymes, metabolites, transporters, and hepatointestinal circulation, coupled with patient-specific genetic profiles and clinical risk factors, complicates the issue. The journey toward personalized CPT-11 chemotherapy is ongoing, yet the drug’s role in treating advanced solid tumors is indispensable.

In the contemporary era of targeted cancer therapy and the surge of immunobiological treatments, it is somewhat paradoxical that cytotoxic agents like CPT-11 continue to be indispensable in oncology. The imperative for clinicians is to optimize these treatments, leveraging medical informatics and cutting-edge science for the ultimate benefit of patients. As previously discussed, preclinical and clinical studies suggest that traditional Chinese medicine (TCM) formulations, herbal prescriptions, and their active constituents may hold potential in preventing or mitigating the symptoms of chronic diarrhea associated with CPT-11-based chemotherapy. The mechanisms by which TCM exerts its effects are multifaceted, primarily targeting the metabolic pathways of CPT-11 through interactions with specific enzymes, metabolites, and transporters. While TCM offers certain advantages in managing CID, the complexity and heterogeneity of its components necessitate further research to elucidate the active ingredients and underlying mechanisms [173]. The integration of novel drug delivery systems with TCM-derived active ingredients may represent a promising frontier in the treatment of CID.

Regardless of the therapeutic approach, the overarching goal is to bolster the tumor’s response to CPT-11, effectively mitigate CID, elevate patient quality of life, and curtail medical expenditures. There is an imperative for sustained research into the etiology of CID and the refinement of targeted treatment strategies. Additionally, vigilant polypharmacy management and the judicious application of CPT-11 in specific clinical contexts are pivotal in reducing the prevalence of CID.

In summary, while CPT-11 remains a cornerstone in cancer treatment, managing its associated diarrhea is a multifaceted challenge. Both conventional and traditional approaches have shown promise, but each comes with its own set of limitations. Future research should focus on developing personalized treatment strategies that integrate the best of both worlds, ensuring optimal patient outcomes.

## Figures and Tables

**Figure 1 pharmaceuticals-18-00359-f001:**
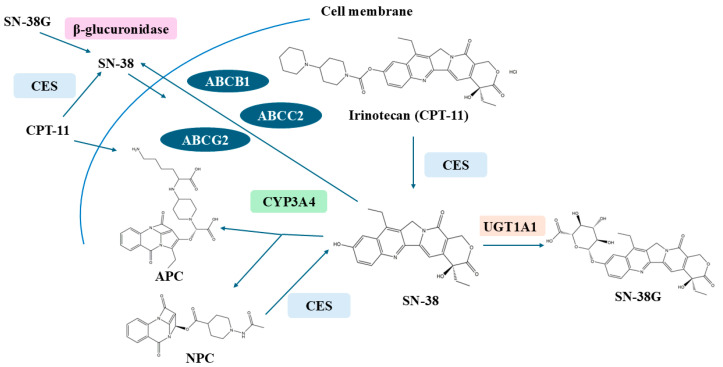
Structure and conformation of irinotecan. The main enzymes implicated in the conversion of irinotecan into its active metabolite SN-38 and the inactive product SN-38G are indicated.

**Figure 3 pharmaceuticals-18-00359-f003:**
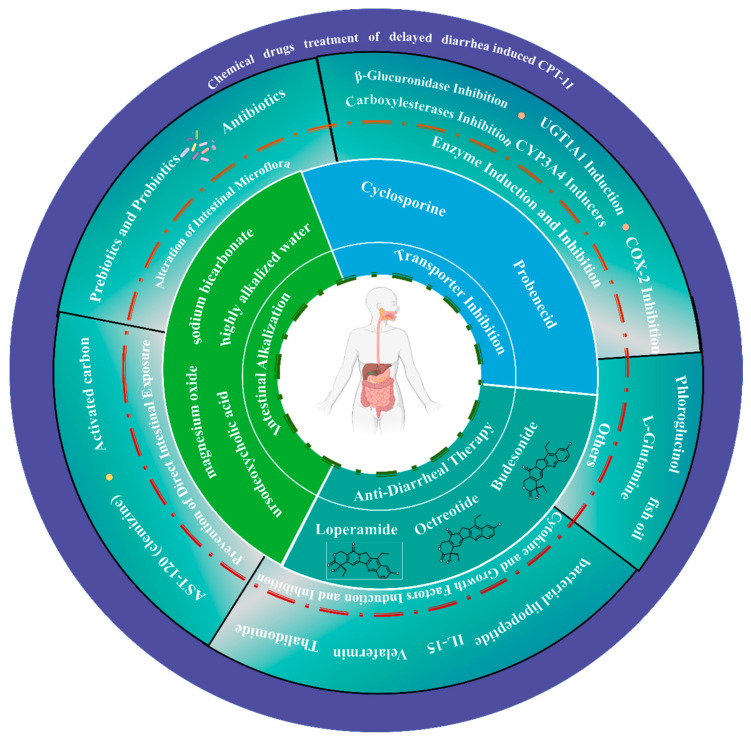
The major chemotherapeutic agents for the treatment of CPT-11-induced diarrhea.

**Figure 4 pharmaceuticals-18-00359-f004:**
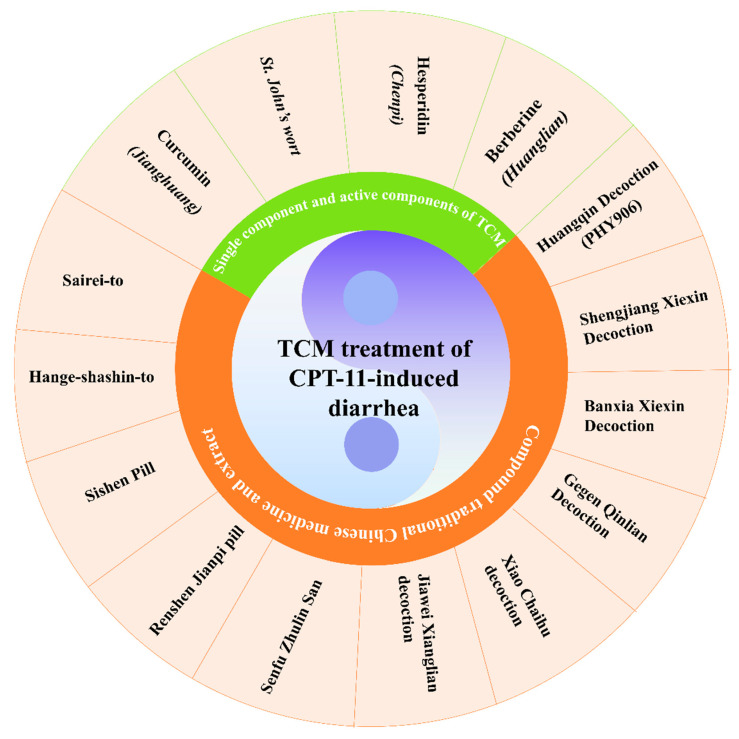
The major traditional Chinese medicine treatment of delayed diarrhea.

**Figure 5 pharmaceuticals-18-00359-f005:**
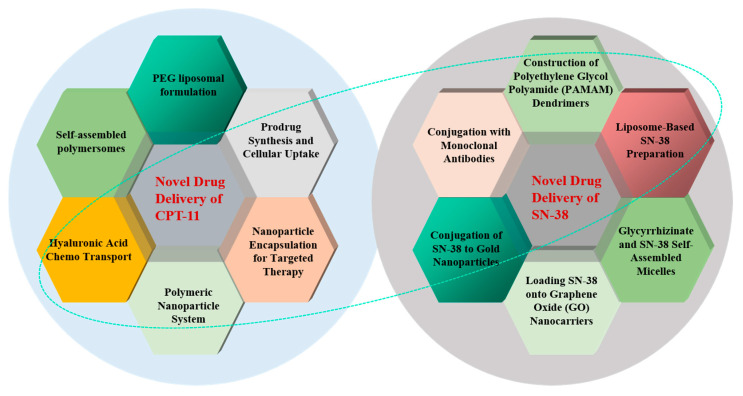
Structural/Chemical modification and novel drug delivery methods.

**Table 1 pharmaceuticals-18-00359-t001:** Comparative analysis of the major therapeutic approaches.

Approach	Strengths	Limitations
Conventional Treatments (Loperamide, Octreotide)	- Well-established in clinical practice - Immediate symptom relief	- High failure rate in severe cases - Potential side effects (e.g., constipation, hyperglycemia)
Traditional Chinese Medicine (TCM)	- Holistic approach - Potential for long - Term benefits - Modulation of gut microbiota	- Standardization challenges - Limited long-term clinical data - Potential drug interactions
Novel Drug Delivery Systems	- Enhanced efficacy - Reduced toxicity - Targeted delivery	- Limited clinical validation - Manufacturing complexities - High costs

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
