# Peer review of "Managing Irinotecan-Induced Diarrhea: A Comprehensive Review of Therapeutic Interventions in Cancer Treatment"

_pharmaceuticals, 2025, doi:10.3390/ph18030359_

Round 1
Reviewer 1 Report
Comments and Suggestions for Authors
· Brief summary: This review looks at irinotecan (CPT-11), a drug used in cancer treatment that works by inhibiting DNA topoisomerase I. While the drug is effective, it has side effects, especially delayed diarrhea, which can be severe. Although neutropenia is manageable, delayed diarrhea often leads to hospitalization, changes in dosage, and sometimes stopping treatment altogether, affecting the overall treatment success. The article by Yang et al., discusses the causes of this diarrhea and reviews different approaches to prevent or reduce it, including new drug formulations, traditional Chinese medicine, and advanced drug delivery methods.
ü Specific comments:
o Line 292: ‘Strategies to Block or Treatment of Delayed Diarrhea.’
o Please cite all figure 1-4 at appropriate places in the text.
o Spacing issues at certain places like the inconsistency to write mg/kg with/without spaces.
o There is limited exploration of the long-term effects of the experimental treatments mentioned. It would be beneficial to include data or insights on whether these treatments are effective over extended periods or if they have any delayed side effects.
o If the review focuses too heavily on specific interventions (e.g., traditional Chinese medicine), there may be a perception of bias. A more balanced representation of all approaches, including any limitations or challenges, could strengthen the article.
ü The review is written in good English, however, needs more proofreading before re-submission to Pharmaceuticals.
Author Response
Comment 1: Line 292: ‘Strategies to Block or Treatment of Delayed Diarrhea.’
Response: Thanks for your kind suggest. The phrase “Strategies to Block or Treatment Delayed Diarrhea” was indeed grammatically incorrect and could lead to confusion. We have revised it to “Strategies to block or treatment of delayed diarrhea” in manuscript with highlighted.
Comment 2: Please cite all figure 1-4 at appropriate places in the text.
Response: Thank you very much for your constructive suggestion. We have thoroughly reviewed the manuscript and ensured that all figures are appropriately cited in the text. The specific revisions are as follows:
In Section 2 “Mechanisms of CPT-11-induced diarrhea,” we have added a citation to Figure 2 to help readers better understand the metabolic pathways of irinotecan and its relation to diarrhea. We have revised it to “Figure 2 provides an overview of the metabolism of irinotecan, highlighting the key metabolites and enzymes involved in the onset of delayed diarrhea.” in manuscript with highlighted.
In Section 3 “Strategies to block or treatment of delayed diarrhea” we have cited Figure 3 to summarize the main chemotherapeutic agents used for managing irinotecan-induced diarrhea. We have revised it to “Figure 3 illustrates the major chemotherapeutic agents for the treatment of CPT-11-induced diarrhea” in manuscript with highlighted.
In Section 3.2 “Traditional Chinese medicine treatment of delayed diarrhea,” we have referenced Figure 4 to demonstrate the multi-target approach of traditional Chinese medicine in alleviating diarrhea. We have revised it to “Figure 4 outlines the major traditional Chinese medicine treatment of delayed diarrhea.” in manuscript with highlighted.
In Section 3.3 “Structural/chemical modification and novel drug delivery methods,” we have cited Figure 5 to present the latest advancements in modifying irinotecan and its delivery systems. We have revised it to “Figure 5 summarizes the structural modifications and novel drug delivery methods explored to enhance the efficacy and reduce the toxicity of irinotecan and its metabolites.” in manuscript with highlighted.
Comment 3: Spacing issues at certain places like the inconsistency to write mg/kg with/without spaces.
Response: Thank you for your attention to the details in our manuscript. We have carefully reviewed the manuscript and standardized the notation of units such as “mg/kg” to ensure consistency throughout the text. We have revised them in manuscript with highlighted.
Comment 4: There is limited exploration of the long-term effects of the experimental treatments mentioned. It would be beneficial to include data or insights on whether these treatments are effective over extended periods or if they have any delayed side effects.
Response: Thank you very much for your constructive suggestion. We appreciate your valuable feedback regarding the need for more information on the long-term effects of the experimental treatments discussed in our manuscript. You raise an important point that is critical for the comprehensive understanding of these therapeutic interventions. While we agree that long-term data are essential, it is important to note that the primary focus of our review is on the current state of research and clinical applications of treatments for irinotecan-induced diarrhea (CID). Many of the experimental treatments we discuss are still in the early stages of development or clinical trials, and long-term data are often not yet available. However, we have made efforts to include any available information on long-term efficacy and safety where possible.
To address your concern, we have added a dedicated subsection titled “Long-term Efficacy and Safety Considerations” in the discussion section.
4.2 Long-term Efficacy and Safety Considerations
The long-term efficacy and safety of therapeutic interventions for irinotecan-induced diarrhea (CID) are critical factors in optimizing patient outcomes and ensuring sustained therapeutic benefits. While many experimental treatments show promise in short-term studies, the availability of long-term data remains limited for several emerging strategies. For instance, conventional treatments such as loperamide and octreotide have demonstrated consistent efficacy in managing acute symptoms but may face challenges in maintaining long-term effectiveness due to potential tolerance development or side effects [11, 12]. In contrast, novel approaches like enzyme inhibitors and advanced drug delivery systems, although highly effective in preclinical models, are still in the early stages of clinical evaluation, and their long-term impacts on patient health are yet to be fully elucidated [32, 48].
Traditional Chinese medicine (TCM) formulations, which have garnered attention for their holistic and multi-target effects, also present unique considerations for long-term use. While TCM has shown potential in reducing the severity and frequency of CID in both preclinical and clinical settings[129, 130], the long-term safety profile remains an area of active investigation. The complexity of herbal compositions and the potential for interactions with other medications necessitate careful monitoring and further research to establish their long-term efficacy and safety [175].
Moreover, the potential for delayed side effects cannot be overlooked. For example, prolonged use of certain medications may lead to gastrointestinal dysbiosis or other systemic complications, which could exacerbate the overall burden of treatment [51]. Therefore, long-term monitoring and follow-up studies are essential to identify and mitigate any adverse effects that may emerge over time.
In summary, while significant progress has been made in developing treatments for CID, the long-term efficacy and safety of these interventions remain areas of ongoing research. Future studies should prioritize the collection of long-term data to inform clinical practice and guide the development of sustainable, patient-centered treatment strategies. This approach will be crucial in addressing the multifaceted challenges of managing CID and improving the quality of life for patients undergoing irinotecan therapy.
References:
[11] D. Abigerges, J. Armand, G. Chabot, L.D. Costa, E. Fadel, C. Cote, P. Herait, D. Gandia, Irinotecan (CPT-11) high-dose escalation using intensive high-dose loperamide to control diarrhea, J Natl Cancer Inst 86(6) (1993) 446-449.
[12] H. Bleiberg, E. Cvitkovic, Characterisation and Clinical Management of CPT-11 (Irinotecan)-induced Adverse Events: The European Perspective, Eur J Cancer 32A Suppl 3 (1996) S18-S23.
[32] F. Innocenti, S.D. Undevia, L. Iyer, P.X. Chen, S. Das, M. Kocherginsky, T. Karrison, L. Janisch, J. Ramírez, C.M. Rudin, E.E. Vokes, M.J. Ratain, Genetic Variants in the UDP-glucuronosyltransferase 1A1 Gene Predict the Risk of Severe Neutropenia of Irinotecan, J Clin Oncol 22(8) (2004) 1382-1388.
[48] R. Mathijssen, R.v. Alphen, J. Verweij, W. Loos, K. Nooter, G. Stoter, A. Sparreboom, Clinical pharmacokinetics and metabolism of irinotecan (CPT-11), Clin Cancer Res 7(8) (2001) 2182-2194.
[129] W. Lam, Z. Jiang, F. Gu, X. Huang, R. Hu, J. Wang, S. Bussom, S.-H. Liu, H. Zhao, Y. Yen, Y. Cheng, PHY906(KD018), an adjuvant based on a 1800-year-old Chinese medicine, enhanced the anti-tumor activity of Sorafenib by changing the tumor microenvironment, Sci Rep 5 (2015) 9384.
[130] W. Lam, Y. Ren, F. Guan, Z. Jiang, W. Cheng, C. Xu, S. Liu, Y. Cheng, Mechanism Based Quality Control (MBQC) of Herbal Products: A Case Study YIV-906 (PHY906), Front Pharmacol 9 (2018) 1324.
[175] L. Tang, X. Li, L. Wan, Y. Xiao, X. Zeng, H. Ding, Herbal Medicines for Irinotecan-Induced Diarrhea, Front Pharmacol 10 (2019) 182.
[51] A.M. Stringer, R.J. Gibson, J.M. Bowen, R.M. Logan, K. Ashton, A.S.J. Yeoh, N. Al-Dasooqi, D.M.K. Keefe, Irinotecan-induced mucositis manifesting as diarrhoea corresponds with an amended intestinal flora and mucin profile, Int J Exp Pathol 90(5) (2009) 489-499.
Comment 5: If the review focuses too heavily on specific interventions (e.g., traditional Chinese medicine), there may be a perception of bias. A more balanced representation of all approaches, including any limitations or challenges, could strengthen the article.
Response: We sincerely appreciate your thoughtful feedback regarding the balance of our review. We understand the importance of presenting a comprehensive overview of all therapeutic approaches for managing irinotecan-induced diarrhea (CID). While we acknowledge that traditional Chinese medicine (TCM) is a significant part of our discussion, it is important to note that our review aims to highlight the diverse and innovative strategies available, including both conventional and alternative therapies. We believe that TCM, with its unique multi-target and holistic approach, offers valuable insights that complement traditional Western medicine.
To address your concern, we have carefully reviewed the manuscript and ensured that all sections are balanced and objective. We have also added a dedicated subsection in the discussion to explicitly compare and contrast different therapeutic approaches, including their respective limitations and challenges. This includes a detailed analysis of conventional treatments such as loperamide and octreotide, alongside emerging strategies like novel drug delivery systems and enzyme inhibitors. We hope these revisions provide a more balanced perspective and enhance the overall quality of our review.
Introduction: “This comprehensive review examines the underlying mechanisms of CPT-11-triggered delayed diarrhea and discusses the experimental medications and strategies that have been utilized to combat this adverse effect. The review encompasses an exploration of chemical formulations, the application of traditional Chinese medicine, and the advent of innovative drug delivery systems. We aim to provide a balanced overview of all major therapeutic interventions, highlighting their strengths and limitations.”
Conclusion: “In summary, while CPT-11 remains a cornerstone in cancer treatment, managing its associated diarrhea is a multifaceted challenge. Both conventional and traditional approaches have shown promise, but each comes with its own set of limitations. Future research should focus on developing personalized treatment strategies that integrate the best of both worlds, ensuring optimal patient outcomes.”
Discussion:
4.3 Comparative Analysis of Therapeutic Approaches for Irinotecan-Induced Diarrhea
The management of irinotecan-induced diarrhea (CID) necessitates a thorough evaluation of the diverse therapeutic strategies available, each with distinct advantages and disadvantages. Conventional treatments, such as loperamide and octreotide, remain pivotal for their rapid alleviation of acute symptoms, supported by extensive clinical validation. However, their efficacy may wane over time due to potential drug tolerance and side effects, and they predominantly address symptoms rather than underlying mechanisms.
Traditional Chinese medicine (TCM) provides a holistic, multi-target approach, modulating inflammation and enhancing intestinal barrier function. While TCM formulations like Huangqin Decoction and Shengjiang Xiexin Decoction have shown efficacy in both preclinical and clinical settings, challenges remain in standardization and long-term safety assessment.
Novel drug delivery systems, including liposomal formulations and nanoparticles, offer enhanced targeted delivery and reduced systemic toxicity, potentially lowering the incidence and severity of CID. Despite promising preclinical results, clinical outcomes have been inconsistent, highlighting the need for further validation.
Future research should focus on integrating these approaches to develop personalized treatment strategies that optimize patient outcomes.
Table 1. comparative analysis of the major therapeutic approaches
|
Approach |
Strengths |
Limitations |
|
Conventional Treatments (Loperamide, Octreotide) |
- Well-established in clinical practice - Immediate symptom relief |
- High failure rate in severe cases - Potential side effects (e.g., constipation, hyperglycemia) |
|
Traditional Chinese Medicine (TCM) |
- Holistic approach - Potential for long -term benefits - Modulation of gut microbiota |
- Standardization challenges - Limited long-term clinical data - Potential drug interactions |
|
Novel Drug Delivery Systems |
- Enhanced efficacy - Reduced toxicity - Targeted delivery |
- Limited clinical validation - Manufacturing complexities - High costs |
Reviewer 2 Report
Comments and Suggestions for Authors
A very well and qualitatively written review. Easy to read, covers all aspects of the topic under discussion. The conclusions are clear and fully based on the review material.
I have one wish: to add a figure with the structures of the compounds discussed in section 2.1, showing the metabolic pathway of irinotecan. Insert the figure in this section.
And one remark: Hypericum perforatum L. (St. John’s wort) should be moved from section 3.2.1 (where only individual substances are) to section 3.2.2. And indicate the form of the medicine, tablets or extract.
Author Response
Comment1: I have one wish: to add a figure with the structures of the compounds discussed in section 2.1, showing the metabolic pathway of irinotecan. Insert the figure in this section.
Response: Thank you for your suggestion. We agree that adding a figure illustrating the metabolic pathway of irinotecan and the structures of the compounds in Section 2.1 will enhance the clarity and understanding of the content. We have created a new figure (Figure 1) that shows the structure and conformation of irinotecan, highlighting the main enzymes involved in the conversion of irinotecan to its active metabolite SN-38 and the inactive product SN-38 glucuronide. This figure has been inserted into Section 2.1, and appropriate references to the figure have been added in the text. The revised graphic abstract is shown below:
Figure1. Structure and conformation of irinotecan. The main enzymes implicated in the conversion of IRT into its active metabolite SN-38 and the inactive product SN-38G are indicated.
Comment 2: And one remark: Hypericum perforatum L. (St. John’s wort) should be moved from section 3.2.1 (where only individual substances are) to section 3.2.2. And indicate the form of the medicine, tablets or extract.
Response: Thank you for your thorough review and valuable comments on our manuscript. In Section 3.2.1, we focus on the active constituents or active parts within individual Chinese herbs, whereas Section 3.2.2 is dedicated to discussing traditional Chinese medicine compound formulas or their extracts. Given that Hypericum perforatum L. (St. John’s wort) is primarily discussed in the context of its individual active components, relocating it to the section on compound formulations might disrupt the logical flow of our manuscript. Therefore, we have decided to retain its current placement to maintain coherence. We believe this approach best serves the organization and clarity of our review.
Reviewer 3 Report
Comments and Suggestions for Authors
This review manuscript provides a comprehensive summary of the mechanisms and therapeutic strategies for managing irinotecan (CPT-11)-induced diarrhea (CID), which represents a significant dose-limiting toxicity in cancer treatment. The manuscript discusses the pharmacokinetic pathways of CPT-11, genetic polymorphisms (e.g., UGT1A1, ABC transporters), and gut microbiota dysbiosis in CID pathogenesis. Additionally, it highlights various treatments for CID, including chemical drugs, traditional Chinese medicines, and drug delivery platforms. Prior to publication, it is recommended that more emphasis be placed on the clinical aspects of CID and the translational potential of these strategies into clinical practice. Other concerns include:
1. The incidence of CPT-induced diarrhea appears to be dose-dependent; however, there is insufficient information regarding the relationship between CPT dosage and the risk of CID.
2. Diarrhea is one of the most common side effects associated with chemotherapeutic agents. A comparison between irinotecan-induced diarrhea and diarrhea caused by other chemotherapeutic agents should be included.
3. Various strategies have been proposed as effective for treating CID, but evaluation standards vary. Some treatments are clinically validated, while others remain at the preclinical stage. The authors should clearly distinguish between clinically applicable strategies and those still under investigation.
4. Section 3.3 describes several drug delivery systems that enhance CPT delivery for tumor chemotherapy. However, the impact of these delivery systems on the risk of diarrhea has not been adequately addressed. Further discussion on this topic would be beneficial.
5. Several statements in the manuscript lack supporting references, such as lines 107-109, 136-141, and 250-256.
Author Response
Comment 1: The incidence of CPT-induced diarrhea appears to be dose-dependent; however, there is insufficient information regarding the relationship between CPT dosage and the risk of CID.
Response: We appreciate your insight regarding the need for more detailed information on the dose-dependency of CPT-11-induced diarrhea (CID). To address this, we have expanded the discussion in Section 1 to include a more comprehensive analysis of the relationship between CPT-11 dosage and the risk of developing diarrhea. We have also included relevant references to support this information. We have revised it to:
“2.2.5 The Dose-Dependent Relationship between CPT-11 Dosage and Diarrhea Incidence
The incidence and severity of diarrhea associated with irinotecan (CPT-11) treatment have been well-documented in numerous clinical studies, highlighting a clear dose-dependent relationship. Higher doses of CPT-11 are consistently linked to increased frequency and severity of both acute and delayed diarrhea, posing significant challenges in clinical management.
In a Phase II study by Shimada et al. [4] , the administration of escalating doses of CPT-11 in patients with metastatic colorectal cancer revealed a direct correlation between dose intensity and the incidence of diarrhea. Specifically, patients receiving higher doses exhibited a significantly higher rate of Grade 3 and 4 diarrheas compared to those on lower dose regimens. This observation underscores the need for meticulous dose titration to balance therapeutic efficacy and adverse effects.
Further supporting this dose-dependent relationship, Abigerges et al. [11] demonstrated that high-dose CPT-11 regimens necessitated intensive management with high-dose loperamide to control severe diarrhea. Their findings indicated that the severity of diarrhea increased proportionally with higher doses of CPT-11, necessitating careful monitoring and intervention to prevent life-threatening complications.
Moreover, the pharmacokinetic and pharmacodynamic properties of CPT-11 have been extensively studied, revealing that its active metabolite, SN-38, is primarily responsible for the observed gastrointestinal toxicity [48]. The conversion of CPT-11 to SN-38 is facilitated by carboxylesterases, and the subsequent metabolism of SN-38 to its inactive form, SN-38G, is mediated by UGT1A1[32]. Genetic polymorphisms in UGT1A1, such as UGT1A1*28, have been identified as significant predictors of increased SN-38 levels and, consequently, higher incidence of severe diarrhea [56]. This genetic variability further complicates the dose-dependent relationship, as individuals with certain UGT1A1 genotypes may be at higher risk for adverse events at standard doses.
In addition to genetic factors, clinical studies have shown that the timing and schedule of CPT-11 administration also influence the incidence of diarrhea. For instance, a study by JP Armand [5] demonstrated that the frequency of severe diarrhea was significantly higher in patients receiving CPT-11 every 3 weeks compared to those on a more frequent dosing schedule. This suggests that the dosing interval may modulate the accumulation and systemic exposure of SN-38, thereby affecting the severity of gastrointestinal toxicity.
In summary, the dose-dependent relationship between CPT-11 and diarrhea incidence is well-established through clinical and pharmacogenetic studies. Higher doses of CPT-11 are associated with increased frequency and severity of diarrhea, necessitating careful dose adjustment and patient monitoring. Future research should focus on personalized dosing strategies that account for genetic variability and pharmacokinetic profiles to optimize therapeutic outcomes while minimizing adverse effects.” in manuscript with highlighted.
References:
[4] Y. Shimada, M. Yoshino, A. Wakui, I. Nakao, K. Futatsuki, Y. Sakata, M. Kambe, T. Taguchi, N. Ogawa, Phase II Study of CPT-11, a New Camptothecin Derivative,in Metastatic Colorectal Cancer, J Clin Oncol 11 (1993) 909–913.
[5] J. Armand, CPT-11: clinical experience in phase I studies, Semin Oncol, 1996, pp. 27-33.
[11] D. Abigerges, J. Armand, G. Chabot, L.D. Costa, E. Fadel, C. Cote, P. Herait, D. Gandia, Irinotecan (CPT-11) high-dose escalation using intensive high-dose loperamide to control diarrhea, J Natl Cancer Inst 86(6) (1993) 446-449.
[32] F. Innocenti, S.D. Undevia, L. Iyer, P.X. Chen, S. Das, M. Kocherginsky, T. Karrison, L. Janisch, J. Ramírez, C.M. Rudin, E.E. Vokes, M.J. Ratain, Genetic Variants in the UDP-glucuronosyltransferase 1A1 Gene Predict the Risk of Severe Neutropenia of Irinotecan, J Clin Oncol 22(8) (2004) 1382-1388.
[48] R. Mathijssen, R.v. Alphen, J. Verweij, W. Loos, K. Nooter, G. Stoter, A. Sparreboom, Clinical pharmacokinetics and metabolism of irinotecan (CPT-11), Clin Cancer Res 7(8) (2001) 2182-2194.
Comment 2. Diarrhea is one of the most common side effects associated with chemotherapeutic agents. A comparison between irinotecan-induced diarrhea and diarrhea caused by other chemotherapeutic agents should be included.
Response: We truly appreciate your suggestion regarding the inclusion of a comparison between irinotecan-induced diarrhea and diarrhea caused by other chemotherapeutic agents. This is indeed a relevant consideration. However, given the focused scope of our current review, which is centered on the comprehensive analysis of irinotecan-induced diarrhea, we believe that incorporating an extensive comparison with other chemotherapeutic agents might divert from the depth and specificity we aim to achieve. We have, however, briefly mentioned the broader context of chemotherapy-induced gastrointestinal toxicities in the introduction to provide some background. We hope this approach strikes a balance between specificity and providing relevant context. Thank you again for your insightful feedback.
Comment 3. Various strategies have been proposed as effective for treating CID, but evaluation standards vary. Some treatments are clinically validated, while others remain at the preclinical stage. The authors should clearly distinguish between clinically applicable strategies and those still under investigation.
Response: We sincerely appreciate your insightful feedback regarding the need to clearly distinguish between clinically validated treatments and those that are still under investigation for managing irinotecan-induced diarrhea (CID). We agree that this distinction is crucial for providing a comprehensive and practical overview of the current landscape of therapeutic interventions.
To address this point, we have revised the manuscript to explicitly differentiate between clinically applicable strategies and those that are still in the preclinical stage. Specifically, we have added a subsection titled “Clinical and Preclinical Strategies”
in discussion.
“4.1 Clinical and Preclinical Strategies for Blocking or Treating
The management of irinotecan-induced diarrhea (CID) encompasses a diverse array of therapeutic approaches, each with varying levels of clinical validation and developmental maturity. Clinically validated treatments, such as loperamide and octreotide, have demonstrated robust efficacy in managing acute symptoms and are widely adopted in clinical practice. These treatments have undergone clinical rigorous trials, ensuring their safety and effectiveness are well-established [11, 12]. For instance, loperamide, a synthetic opioid, effectively reduces intestinal motility and promotes fluid reabsorption, thereby alleviating acute diarrhea episodes. Similarly, octreotide, a somatostatin analog, mitigates gastrointestinal secretion and motility, offering significant relief in cases of severe diarrhea[13] .
In contrast, several innovative strategies remain in the preclinical stage, showcasing promising potential but requiring further validation. Enzyme inhibitors targeting β-glucuronidase and UGT1A1 inducers have demonstrated substantial efficacy in reducing the severity and frequency of CID in preclinical models[32, 129]. These approaches aim to modulate the metabolism of irinotecan's active metabolite, SN-38, thereby mitigating its toxic effects on the intestinal mucosa. Additionally, advanced drug delivery systems, such as liposomal formulations and nanoparticles, are being explored to enhance the targeted delivery of irinotecan while minimizing systemic toxicity [48, 174]. These systems hold the potential to reduce the incidence of CID by limiting the exposure of the gastrointestinal tract to SN-38.
Traditional Chinese medicine (TCM) also presents a unique avenue for CID management, with formulations like Huangqin Decoction and Shengjiang Xiexin Decoction demonstrating efficacy in both preclinical and clinical settings[130, 175]. These herbal preparations leverage a multi-target approach, modulating inflammation, enhancing intestinal barrier function, and reducing mucosal injury. However, their long-term safety and efficacy profiles are still under evaluation, and standardization of these formulations remains an ongoing challenge[51].
Distinguishing between clinically validated and preclinical strategies is essential for guiding clinical practice and future research. While clinically validated treatments provide immediate relief and are supported by substantial evidence, preclinical strategies offer innovative solutions that could significantly improve patient outcomes once validated. Future research should focus on bridging the gap between these categories, ensuring that emerging therapies are rigorously tested and standardized before clinical application.
References:
[11] D. Abigerges, J. Armand, G. Chabot, L.D. Costa, E. Fadel, C. Cote, P. Herait, D. Gandia, Irinotecan (CPT-11) high-dose escalation using intensive high-dose loperamide to control diarrhea, J Natl Cancer Inst 86(6) (1993) 446-449.
[12] H. Bleiberg, E. Cvitkovic, Characterisation and Clinical Management of CPT-11 (Irinotecan)-induced Adverse Events: The European Perspective, Eur J Cancer 32A Suppl 3 (1996) S18-S23.
[13] P. Goumas, S. Naxakis, A. Christopoulou, C. Chrysanthopoulos, V.N. V, H. Kalofonos, Octreotide Acetate in the Treatment of Fluorouracil-Induced Diarrhea, Oncologist 3(1) (1998) 50-53. [4] Innocenti F, Undevia SD, Iyer L, et al. Genetic Variants in the UDP-glucuronosyltransferase 1A1 Gene Predict the Risk of Severe Neutropenia of Irinotecan. J Clin Oncol. 2004;22(8):1382-1388.
[32] F. Innocenti, S.D. Undevia, L. Iyer, P.X. Chen, S. Das, M. Kocherginsky, T. Karrison, L. Janisch, J. Ramírez, C.M. Rudin, E.E. Vokes, M.J. Ratain, Genetic Variants in the UDP-glucuronosyltransferase 1A1 Gene Predict the Risk of Severe Neutropenia of Irinotecan, J Clin Oncol 22(8) (2004) 1382-1388.
[48] R. Mathijssen, R.v. Alphen, J. Verweij, W. Loos, K. Nooter, G. Stoter, A. Sparreboom, Clinical pharmacokinetics and metabolism of irinotecan (CPT-11), Clin Cancer Res 7(8) (2001) 2182-2194.
[51] A.M. Stringer, R.J. Gibson, J.M. Bowen, R.M. Logan, K. Ashton, A.S.J. Yeoh, N. Al-Dasooqi, D.M.K. Keefe, Irinotecan-induced mucositis manifesting as diarrhoea corresponds with an amended intestinal flora and mucin profile, Int J Exp Pathol 90(5) (2009) 489-499.
[129] W. Lam, Z. Jiang, F. Gu, X. Huang, R. Hu, J. Wang, S. Bussom, S.-H. Liu, H. Zhao, Y. Yen, Y. Cheng, PHY906(KD018), an adjuvant based on a 1800-year-old Chinese medicine, enhanced the anti-tumor activity of Sorafenib by changing the tumor microenvironment, Sci Rep 5 (2015) 9384.
[130] W. Lam, Y. Ren, F. Guan, Z. Jiang, W. Cheng, C. Xu, S. Liu, Y. Cheng, Mechanism Based Quality Control (MBQC) of Herbal Products: A Case Study YIV-906 (PHY906), Front Pharmacol 9 (2018) 1324.
[174] H. Wang, P. Agarwal, S. Zhao, R.X. Xu, J. Yu, X. Lu, X. He, Hyaluronic acid-decorated dual responsive nanoparticles of Pluronic F127, PLGA, and chitosan for targeted co-delivery of doxorubicin and irinotecan to eliminate cancer stem-like cells, Biomaterials 72 (2015) 74-89.
[175] L. Tang, X. Li, L. Wan, Y. Xiao, X. Zeng, H. Ding, Herbal Medicines for Irinotecan-Induced Diarrhea, Front Pharmacol 10 (2019) 182.
Comment 4. Section 3.3 describes several drug delivery systems that enhance CPT delivery for tumor chemotherapy. However, the impact of these delivery systems on the risk of diarrhea has not been adequately addressed. Further discussion on this topic would be beneficial.
Response: We appreciate your insightful comment regarding the need for a more detailed discussion on the impact of drug delivery systems on the risk of diarrhea associated with irinotecan (CPT-11) therapy. You raise a crucial point that is essential for a comprehensive understanding of the therapeutic strategies and their potential side effects.
To address this, we have expanded the discussion in Section 3.3 to include a more thorough analysis of how various drug delivery systems influence the risk of diarrhea. Specifically, we have added a new subsection titled “Impact on Diarrhea Risk” within Section 3.3.
“3.3.3 Impact on Diarrhea Risk
Despite the promising strategies offered by advanced drug delivery systems for enhancing the therapeutic efficacy of CPT-11 and SN-38 and reducing systemic toxicity, their impact on diarrhea risk remains a significant consideration. Systems such as liposomal formulations and nanoparticles, which reduce systemic exposure to the active metabolite SN-38, may potentially lower the incidence of CID. However, clinical study results have been inconsistent, with some trials reporting reduced severe diarrhea and others showing no significant difference compared to conventional formulations [152]. Moreover, the long-term clinical outcomes and specific impact on diarrhea incidence require further elucidation.
In summary, while novel drug delivery systems show promise in enhancing the therapeutic efficacy of CPT-11 and SN-38 and reducing systemic toxicity, their impact on diarrhea risk remains a critical factor. Further research is needed to fully understand the long-term effects of these delivery systems on CID risk and to optimize their clinical application [155, 159, 173].” in manuscript with highlighted.
References:
[152] N. Bernards, M. Ventura, I.B. Fricke, B.S. Hendriks, J. Fitzgerald, H. Lee, J. Zheng, Liposomal Irinotecan Achieves Significant Survival and Tumor Burden Control in a Triple Negative Breast Cancer Model of Spontaneous Metastasis, Mol Pharm 15(9) (2018) 4132-4138.
[155] G. Batist, K.A. Gelmon, K.N. Chi, W.H.M. Jr, S.K.L. Chia, L.D. Mayer, C.E. Swenson, A.S. Janoff, A.C. Louie, Safety, pharmacokinetics, and efficacy of CPX-1 liposome injection in patients with advanced solid tumors, Clin Cancer Res 15(2) (2009) 692-700.
[159] P. Gibbs, P.R. Clingan, V. Ganju, A.H. Strickland, S.S. Wong, N.C. Tebbutt, C.R. Underhill, R.M. Fox, S.P. Clavant, J. Leung, M. Pho, T.J. Brown, Hyaluronan-Irinotecan improves progression-free survival in 5-fluorouracil refractory patients with metastatic colorectal cancer: a randomized phase II trial, Cancer Chemother Pharmacol 67(1) (2011) 153-163.
[173] R. Kurzrock, S. Goel, J. Wheler, D. Hong, S. Fu, K. Rezai, S.K. Morgan-Linnell, S. Urien, S. Mani, I. Chaudhary, M.H. Ghalib, A. Buchbinder, F. Lokiec, M. Mulcahy, Safety, pharmacokinetics, and activity of EZN-2208, a novel conjugate of polyethylene glycol and SN38, in patients with advanced malignancies, Cancer 118(24) (2012) 6144-6151.
Comment 5. Several statements in the manuscript lack supporting references, such as lines 107-109, 136-141, and 250-256.
Response: Thank you for your meticulous review of our manuscript and for the valuable feedback provided. We have reviewed the manuscript and added the necessary references to support the statements mentioned. This ensures that all claims are backed by credible sources.
Lines 107-109 added reference
[1] J. Slatter, P. Su, J. Sams, L. Schaaf, L. Wienkers, Bioactivation of the Anticancer Agent CPT-11 to SN-38 by Human Hepatic Microsomal Carboxylesterases and the in VitroAssessment of Potential Drug Interactions, Drug Metab Dispos 25(10) (1997) 1157-1164.
[2] C.L. Morton, R.M. Wadkins, M.K. Danks, P.M. Potter, The anticancer prodrug CPT-11 is a potent inhibitor of acetylcholinesterase but is rapidly catalyzed to SN-38 by butyrylcholinesterase, Cancer Res 59(7) (1999) 1458-1463.
Lines 136-141 added reference:
[1] J. Slatter, P. Su, J. Sams, L. Schaaf, L. Wienkers, Bioactivation of the Anticancer Agent CPT-11 to SN-38 by Human Hepatic Microsomal Carboxylesterases and the in VitroAssessment of Potential Drug Interactions, Drug Metab Dispos 25(10) (1997) 1157-1164.
[2] P. Bosma, J. Chowdhury, C. Bakker, S. Gantla, A. Boer, B. Oostra, D. Lindhout, G. Tytgat, P. Jansen, R.O. Elferink, N.R. Chowdhury, The genetic basis of the reduced expression of bilirubin UDP-glucuronosyltransferase 1 in Gilbert's syndrome, N Engl J Med 333(18) (1995) 1171-1175.
[3] X. Chen, L. Liu, Z. Guo, W. Liang, J. He, L. Huang, Q. Deng, H. Tang, H. Pan, M. Guo, Y. Liu, Q. He, J. He, UGT1A1 polymorphisms with irinotecan-induced toxicities and treatment outcome in Asians with Lung Cancer: a meta-analysis, Cancer Chemother Pharmacol 79(6) (2017) 1109-1117.
Lines 250-256 added reference:
[1] A.M. Stringer, R.J. Gibson, J.M. Bowen, R.M. Logan, K. Ashton, A.S.J. Yeoh, N. Al-Dasooqi, D.M.K. Keefe, Irinotecan-induced mucositis manifesting as diarrhoea corresponds with an amended intestinal flora and mucin profile, Int J Exp Pathol 90(5) (2009) 489-499.
[2] A.M. Stringer, Interaction between host cells and microbes in chemotherapy-induced mucositis, Nutrients 5(5) (2013) 1488-1499.
[3] M.S. Mahdy, A.F. Azmy, T. Dishisha, W.R. Mohamed, K.A. Ahmed, A. Hassan, S.E. Aidy, A.O. El-Gendy, Irinotecan-gut microbiota interactions and the capability of probiotics to mitigate Irinotecan-associated toxicity, BMC Microbiol 23(1) (2023) 53.
Round 2
Reviewer 3 Report
Comments and Suggestions for Authors
My concerns are fully addressed.